# Spatial and temporal variability of environmental proxies from the top 120 m of two ice cores in Dronning Maud Land (East Antarctica)

Sarah Wauthy[1], Jean-Louis Tison[1], Mana Inoue[1], Saïda El Amri[1], Sainan Sun[2], François Fripiat[1], Philippe Claeys[3] and Frank Pattyn[1]

[1]Laboratoire de Glaciologie, Université libre de Bruxelles (ULB), Brussels, Belgium
[2]Department of Geography and Environmental Sciences, Northumbria University, Newcastle upon Tyne, UK
[3]Analytical-Environmental and Geochemistry, Vrije Universiteit Brussel (VUB), Brussels, Belgium

*Correspondence to*: Sarah Wauthy (sarah.wauthy@ulb.be)

**Abstract.** The Antarctic ice sheet's future contribution to sea level rise is difficult to predict, mostly because of the uncertainty and variability of the surface mass balance (SMB). Ice cores are used to locally (km scale) reconstruct SMB with a very good temporal resolution (up to sub-annual), especially in coastal areas where accumulation rates are high. The number of ice cores records has been increasing these last years, revealing an important spatial variability and different trends of SMB, highlighting
the crucial need for greater spatial and temporal representativeness.

We present records of density, water stable isotopes ($\delta^{18}$O, $\delta$D and deuterium excess), major ions concentrations ($Na^+$, $K^+$, $Mg^+$, $Ca^+$, MSA, $Cl^-$, $SO_4^{2-}$ and $NO_3^-$), and continuous electrical conductivity measurement (ECM), as well as age models and resulting surface mass balance from the top 120 m of two ice cores (FK17 and TIR18) drilled on two adjacent ice rises located in coastal Dronning Maud Land and dating back to the end of the 18th century. Both environmental proxies and SMB show
contrasting behaviors, suggesting strong spatial and temporal variability at the regional scale. In terms of precipitation proxies, both ice cores show a long-term decrease of deuterium excess (d-excess) and a long-term increase of $\delta^{18}$O, although less pronounced. In terms of chemical proxies, the non-sea-salt sulfate ($nssSO_4^{2-}$) concentrations of FK17 are twice the ones of TIR18 and display an increasing trend on the long-term while there is only a small increase after 1950 in TIR18. The $SO_4^{2-}$/$Na^+$ ratios show a similar contrast between FK17 and TIR18 and are consistently higher than the sea water ratio, indicating a
dominant impact of the $nssSO_4^{2-}$ on the $SO_4^{2-}$ signature. The mean long-term SMB is similar for FK17 and TIR18 ($0.57 \pm 0.05$ and $0.56 \pm 0.05$ m i.e. a$^{-1}$ respectively), but the annual records are very different: since the 1950's, TIR18 shows a continuous decrease while FK17 has shown an increasing trend until 1995 followed by a recent decrease. The datasets presented here offer numerous development possibilities for the interpretation of the different paleo profiles and for addressing the mechanisms behind the spatial and temporal variability observed at the regional scale (tens of km scale) in East Antarctica.
The "Mass2Ant IceCores" datasets are available on Zenodo (https://doi.org/10.5281/zenodo.7848435; Wauthy et al., 2023).

## 1 Introduction

The Antarctic ice sheet's future contribution to global sea level rise, with a potential of 58.0 m, is difficult to predict, partly because of the uncertainty and variability of the surface mass balance (SMB) (Frezzotti et al., 2013; Thomas et al., 2017; IPCC
2019). The surface mass balance is the difference at the ice sheet surface between mass gain, by snowfall, and to a lesser extent through wind deposition, and mass loss, by sublimation, wind-driven ablation and wind-driven sublimation (surface melt has been shown to be negligible across the majority of the ice sheet, Lenaerts et al., 2019).

The amount of snowfall is the largest driver of Antarctic SMB (Van Wessem et al., 2018; Agosta et al., 2019) and is mainly controlled by three mechanisms: thermodynamics, large-scale dynamics and synoptic-scale dynamics (Dalaiden et al., 2020). Thermodynamics refers to the effect of higher temperatures on snowfall: under a warming climate, the atmospheric moisture content is expected to increase, increasing precipitation resulting in higher snow accumulation (Krinner et al., 2007; Palerme et al., 2014). This process could partly mitigate sea level rise. Large-scale atmospheric dynamics refer to the meridional (southward) transport of moisture from lower latitudes (Lenaerts et al., 2019). This results in regional variability as local topography broadly controls the precipitation, with sometimes strong spatial SMB variations within kilometers (Agosta et al., 2012). For example, ice rises are known to influence the local SMB distribution by enhancing precipitation on the windward side (due to orographic uplift) and by influencing the snow erosion on the leeward side of the ice rise (Lenaerts et al., 2014). Synoptic-scale dynamics refer to short-lived events, such as atmospheric rivers that can significantly contribute to local SMB (Gorodetskaya et al., 2013; Maclennan et al., 2022). Atmospheric rivers are long and narrow bands of high atmospheric moisture that protrude from the midlatitudes to the high latitudes and result in snowfall events of large magnitude, like the extreme snowfalls of 2009 and 2011 that resulted in high SMB anomalies in East Antarctica (Lenaerts et al., 2013; Gorodetskaya et al., 2014; Philippe et al., 2016).

In addition to its large spatial variability, the SMB is also highly variable in time with natural variability ranging from (sub-) daily to interannual (and decadal) time scales (Lenaerts et al., 2019). Long-term changes (trends) caused by external forcings, especially the anthropogenic warming, are also identified: an increase in the Antarctic-wide SMB was observed between 1801 and 2000 but the trends are highly variable in magnitude and even in sign at the regional scale (Medley and Thomas, 2019). The Antarctic SMB and its past and present changes therefore need to be better understood to improve predictions of Antarctica's future contribution to sea level rise (Thomas et al., 2017). If different methods, like regional climate models and airborne or spaceborne instrumental data, are currently available to study the SMB and its variability, ice cores provide the only record of SMB before the instrumental and satellite period. They are used to locally reconstruct SMB with a very good temporal resolution (annual to pluri-annual), especially in coastal areas where high accumulation rates allow to study (sub-) annually resolved proxy records.

The number of accumulation records from ice cores has been increasing these last years, revealing an important spatial variability and different trends of accumulation history (Thomas et al., 2017). There are only a few ice core records extending back to more than 200 years but it has been shown that SMB changes over most of Antarctica are statistically negligible over the previous 800 years (Frezzotti et al., 2013), while the four ice core records covering the last 1000 years suggest a decrease in SMB over this period (Thomas et al., 2017). Different trends are also observed regionally on shorter time scales. For example, in coastal Dronning Maud Land (DML), an ice core from an ice rise shows an increasing accumulation trend since the 1950's (Philippe et al., 2016) while the other cores in this coastal region show a negative SMB trend in the recent decades (Kaczmarska et al., 2004; Sinisalo et al., 2013; Schlosser et al., 2014; Altnau et al., 2015; Vega et al., 2016; Ejaz et al., 2021). This highlights the crucial need for a greater spatial and temporal representativeness.

If it is essential to assess the multiscale spatial and temporal variability of SMB from observation data, it is also fundamental to see if it can be reasonably reproduced in regional and global climate models. This is the purpose of the Belgian funded Mass2Ant project (contract # BR/165/A2/Mass2Ant – BELSPO "BRAIN.be") which aims to better understand the processes controlling the surface mass balance in East Antarctica and its variability within the last centuries and test its reproducibility at local, regional and global scale in climate models, in order to improve the projections of mass balance changes for the East Antarctic ice sheet. To this goal, the project is combining different approaches: observations from new ice cores and radar profiles, existing databases, and model simulations.

In this paper, we present density, water stable isotopes ($\delta^{18}$O, $\delta$D and d-excess), ion concentrations (Na$^+$, K$^+$, Mg$^+$, Ca$^+$, MSA, Cl$^-$, SO$_4^{2-}$ and NO$_3^-$), and continuous electrical conductivity measurement (ECM) records, age models, and resulting SMB from the top 120 m of two ice cores drilled on two ice rises located in coastal DML (Fig. 1). We date the ice cores back to the end of the eighteenth century (CE 1793 ± 3 years and 1780 ± 5 years) by layer counting, and calculate SMB after correction for vertical strain rates. We then compare these new records to other coastal ice cores in DML, especially to the Derwael ice rise core (Philippe et al., 2016; Philippe and Tison, 2023) which is located ca. 100 km to the east of our two new ice core sites, and demonstrate the richness of these datasets for further studies addressing the mechanisms behind spatial and temporal variability of SMB and environmental proxies in East Antarctica.

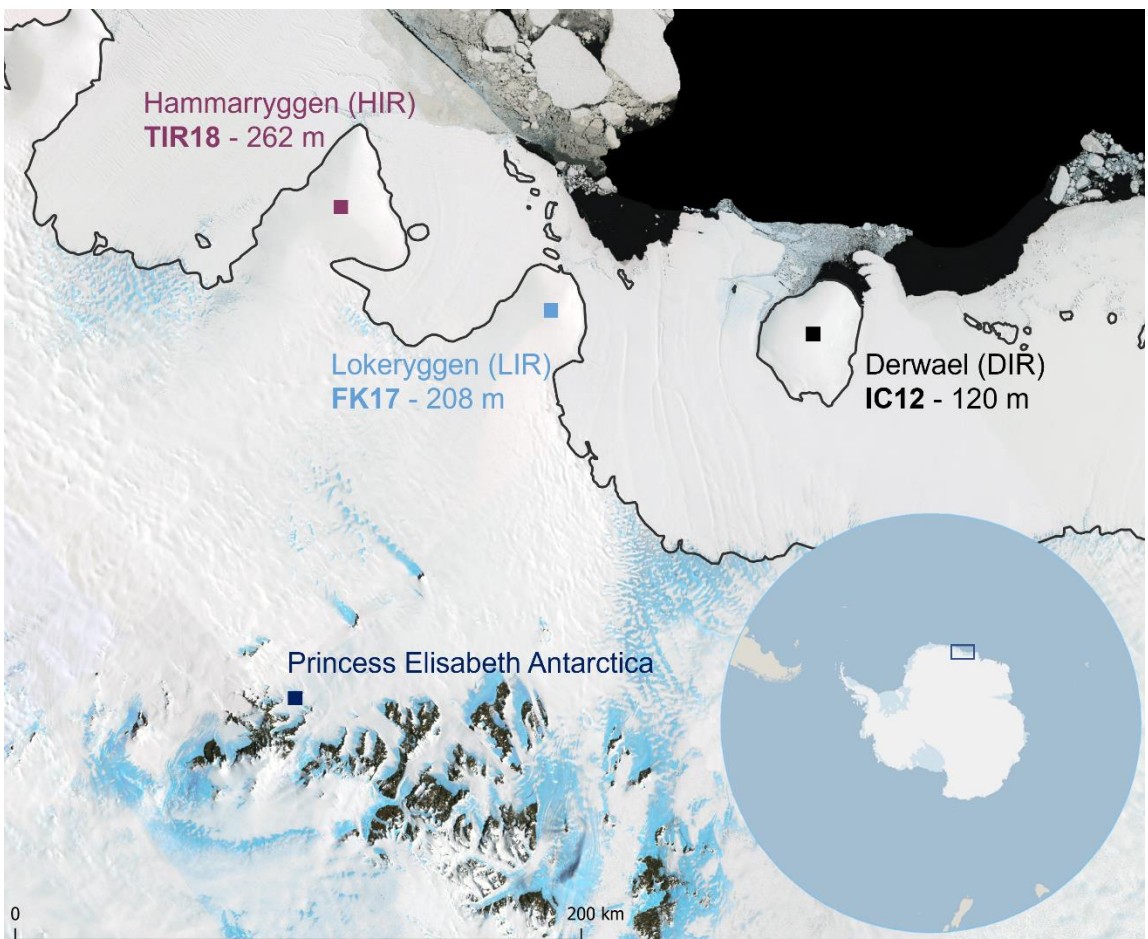

**Figure 1. Location of the two ice cores (TIR18 and FK17) at the Princess Ragnhild Coast (Dronning Maud Land, East Antarctica) with the three ice rises, from west to east: Hammarryggen Ice Rise (HIR), Lokeryggen Ice Rise (LIR) and Derwael Ice Rise (DIR). The SMB of IC12 core (DIR) has been published in Philippe et al., (2016). The station Princess Elisabeth Antarctica, near the Sør Rondane Mountains, is indicated. The grounding line is represented in black. The location of the area is framed in the panel at the bottom right. This figure was prepared with Quantarctica (Matsuoka et al., 2021).**

## 2 Methods

### 2.1 Field

Our study sites are located on the coastal Dronning Maud Land, along the Princess Ragnhild Coast, East Antarctica (Fig. 1). Two ice cores were drilled using an intermediate-depth ice core drill (ECLIPSE, Icefield Instruments, Inc.) at the crest of two adjacent ice rises (Hammarryggen and Lokeryggen). From a geomorphological point of view, these ice rises are ice promontories connected to the grounded ice sheet to the south and surrounded by ice shelves to the east, north and west. Ice rises are preferred drilling sites due to their coastal position (favoring high accumulation) and their own flow system, with

little disturbance by horizontal flow at the crest of the dome (Matsuoka et al., 2015). However, local dynamic conditions at the drilling locations have recently been shown to potentially affect the absolute value of accumulation although not its temporal variability (Cavitte et al., 2022). More generally, ice rises perturb air masses trajectories and induce snowfall with orographic precipitation resulting in higher accumulation on the windward side and less accumulation on the leeward side of the ice divide (Lenaerts et al., 2014).

The drill sites were pre-selected based on the high-resolution REMA (Reference Elevation Model of Antarctica, Howat et al., 2019) to roughly identify the highest point of the ice rise and then selected in the field based on detailed surface topographic measurements using GNSS (Global Navigation Satellite System). The Lokeryggen ice rise (LIR) reaches approximately 333 m above sea level with an ice thickness of ~420 m. The ice core, named FK17, was drilled during the 2017/2018 austral summer (-70.53648° S, 24.07036° E) and is 208 m long (Fig. 1). A ~2 m deep trench was used for the drill set-up so that a shallow ice core (FK18, 9 m) was retrieved the next season to obtain a complete age-depth profile. The cores were directly cut in 50 cm sections in the field. The Hammarryggen ice rise (HIR) reaches approximately 348 m above sea level with an ice thickness of ~550 m. The ice core, named TIR18, was drilled during the 2018/2019 austral summer (-70.49960° S, 21.88017° E) and is 262 m long (Fig. 1). To allow retrieval of good quality cores, the drilling fluid ESTISOL 140 was used from 98 m onwards. A shallow core of 10 m (TIR18-shallow) was retrieved to obtain a complete age-depth profile since a 2.40 m deep trench was required for the drill set-up. The ice cores were directly cut into 50 cm sections in the field and then triple weighted to get depth-density profile. For both drillings, the ice cores were logged, packed, and shipped at -20 °C to the home laboratory for analysis. Note that we focused our analyses on the top 120 meters of each ice core, resulting in approximately 5000 samples to measure for the whole set of variables.

The top 50 meters of the FK17 and TIR18 datasets have already been used to provide preliminary age models in two publications which use ground-penetrating radar data and combine them with ice core information to study the influence of ice rises on surface mass balance (Kausch et al., 2020) and to evaluate the representativeness of ice core derived SMBs (Cavitte et al., 2022). The main conclusions of these papers are presented in section 4.1 to put our results into context and perspective.

Two automatic weather stations (AWSs) were installed on the eastern side and on the western side of the LIR. The AWS on the western side stopped recording after a few months but the AWS on the eastern side recorded nearly two years of sub-daily wind, temperature and snow accumulation. The second AWS was dismantled on the 12th of December 2019 and the records thus lack the end of the year 2019. The AWSs measured an average southeast wind direction of ~ 132 °, indicating that the eastern flanks of the ice rises are the windward side, and the western flanks of the ice rises are the leeward side (Kausch et al., 2020). The temperature and snow accumulation records are shown in Appendix A. The average temperature between the 1st of January and the 12th of December is -16.5 °C in 2018 and –16.6 °C in 2019. The total snow accumulation between January and end of November is 1.11 m in 2018 and 0.68 m in 2019.

## 2.2 Measured data

### 2.2.1 Ice core processing

In the laboratory, the cores were stored in freezers at -25°C before analysis. In the cold room at -20°C, the cores were cut with a clean bandsaw. One half of the core was used for electrical conductivity measurement (ECM) and then kept as an archive. The outer part of the second half was used for water stable isotopes measurement, cut into 5 cm length samples that were melted prior to analysis. The liquid samples were transferred into 4 mL vials, completely filled to prevent contact with air. The inner part of the core was cut into two sticks of 3 cm x 3 cm square section. To analyze major ions, the first stick was decontaminated by removing ~ 2 mm from each face using an ethanol-cleaned microtome blade in a class-100 laminar flow

hood and then cut loose from the stick every 5 cm by striking the stick with the microtome knife. The samples were melted in their clean bottles and poured into clean vials after three rinses in a laminar flow hood. Blank ice samples were used to check for contamination by freezing Milli-Q® water and processing it in the same way as the samples. The second stick was kept as
archive or used for duplicate analysis.

While the density of the 50 cm sections of the TIR18 core was directly and continuously measured in the field, the 50 cm sections from FK17 were discretely measured in the home laboratory. Every 2 to 4 meters, one stick was triple measured, and triple weighted, providing a discontinuous FK17 density profile. The continuous depth-density profiles are obtained following
the equation from Morgan et al., (1998):

$$\rho_z = \rho_{ice} - (\rho_{ice} - \rho_{surf}) \exp(-kz), \qquad\qquad\qquad (1)$$

where $\rho_z$ is the density at depth $z$, $\rho_{ice}$ is the density of glacier ice and equals 917 kg m$^{-3}$, $\rho_{surf}$ is the surface density and $k$ is a constant. Surface density has been shown to be highly variable over LIR and HIR with variations over tens of meters (Wever et al., 2022; Cavitte et al., 2022). A best fit with the measured densities resulted in the attribution of surface density values of
430 kg m$^{-3}$ and 420 kg m$^{-3}$ and constant $k$ values of 0.0301 and 0.0316 for FK17 and TIR18, respectively (Morgan et al., 1998).

### 2.2.2 Sample measurements

The water stable isotopes ($\delta^{18}O$ and $\delta D$ vs. VSMOW) were measured using PICARRO L 2130-i cavity-ring down spectrometer (CRDS). Major ions (Na$^+$, K$^+$, Mg$^+$, Ca$^+$, MSA, Cl$^-$, NO$_3^-$, SO$_4^{2-}$) analysis was performed using a Dionex-ICS5000 liquid chromatography (see details on the method in Appendix Table B1). The weight ratio SO$_4^{2-}$/Na$^+$ was calculated and its
seasonality was used to date the ice cores. The non-sea-salt sulfate (nssSO$_4^{2-}$) record was also calculated as in Abram et al., (2013):

$$(nssSO_4^{2-}) = (SO_4^{2-}) - 0.25 \times (Na^+) \qquad\qquad\qquad (2)$$

It is expressed in ppb and represents all SO$_4^{2-}$ aerosols that are not from marine origin. We normalized nssSO$_4^{2-}$ by subtracting the mean and dividing by the standard deviation to identify the potential volcanic horizons in the ice core.


ECM was also used to identify volcanic horizons. Once the conductivity of the ice has been corrected for temperature, the signal depends mainly on its acidity, which varies seasonally but is also associated to sulfate emissions during volcanic eruptions (Hammer, 1980; Hammer et al., 1994). The Handheld ECM unit V3 was designed and manufactured by Icefield Instrument Inc. ECM was carried out in the cold room at -20°C by applying a direct current (1000 V) on the freshly cleaned
surface of the half core. The resulting signal (4 mm resolution) was corrected for temperature and then for porosity as the cores were principally made of snow and firn. As ECM is inversely proportional to air content, the ECM signal was multiplied by the ratio of the ice density to the snow/firn density using the depth-density profile of the corresponding ice core (Kjaer et al., 2016). The resulting ECM signal was either smoothed to study its seasonality using a 121-points second order Savitzky-Golay filter (Savitzky and Golay, 1964) in order to reduce the noise, or normalized by subtracting the mean and dividing by the
standard deviation to identify volcanic eruptions. Peaks larger than 3σ were considered as potential volcanic horizons.

### 2.2.3 Data quality assessment

The uncertainty of the various datasets has been assessed using either reproducibility (for ECM), error propagation calculation (for density) or long-term reproducibility with internal standards (for water isotopes and ions measurements). The relative standard deviation (RSD) of ECM is estimated at 14 % from triplicate analysis of 20 sections along the cores, which represents
a mean current signal of 7 ± 1 µA. Using only the two first scans brings the RSD down to 8 %, this can be explained by the

space charge polarization effect (Hammer, 1980; Moore et al., 1992). The reproducibility in peak location in single cores sections has been estimated from triplicate measurements to be ± 3 mm, similar to the chosen resolution. The density is calculated as the ratio between repeated measurements of mass and volume and the error on the density is therefore obtained from the error propagation on the mass and volume. The volume is a stick (defined by length and two sides of a square base) for FK17 and a cylinder (defined by length and radius) for TIR18. The measurements of these lengths and radius are associated with a known uncertainty defined by the instrument used. The mass was measured using precise scales with errors of 0.5 g and 1 g, for FK17 and TIR18 respectively. For both density profiles, the uncertainty is estimated to be <3 %. The standard deviation (SD) of the water isotopes is calculated using an internal standard which has been corrected and referenced to the VSMOW scale using three previously calibrated in-house standards. The long-term reproducibility (i.e., SD) is 0.02 ‰ for $\delta^{18}O$, 0.2 ‰ for $\delta D$ and 0.4 ‰ for d-excess. Note that since d-excess results from a calculation based on $\delta^{18}O$ and $\delta D$ (i.e., d-excess = $\delta D - 8 \times \delta^{18}O$, Dansgaard, 1964), the error propagation principle described above is applied to derive its uncertainty. The standard deviation and accuracy of ions concentration is also estimated using two internal standards (one "high" and one "low" concentration) measured two times in each run. The SD and accuracy obtained are presented in Table 1. Note that the measured concentrations are well above the detection limit calculated for each ion.

| | $Na^+$ | $K^+$ | $Mg^+$ | $Ca^{2+}$ | MSA | $Cl^-$ | $SO_4^{2-}$ | $NO_3^-$ |
|---|---|---|---|---|---|---|---|---|
| SD (ppb) | 5 | 1 | 1 | 1 | 1 | 9 | 5 | 2 |
| Accuracy (%) | 3 | 5 | 5 | 5 | 4 | 4 | 4 | 4 |

**Table 1. Standard deviation (SD) and accuracy of the ions concentration defined using two internal standards.**

We also considered potential bias effects of melt layers on our records. During the core logging, clear ice layers were identified as "melt layers". These layers were very thin (< 1 mm) in most cases with few bubble-free lentils and layers of maximum ~5 mm. Given the sample resolution (5 cm) and the thinness of the layers, no significant impact is expected on the measured and derived data. The frequency distributions of $\delta^{18}O$ samples "with" and "without" clear ice layers are shown for FK17 and TIR18 in Appendix C. The medians are relatively close: -18.79 ‰ vs. -18.95 ‰ for the FK17 samples "with" and "without" clear ice layers respectively, and -19.48 ‰ vs. -19.21 ‰ for the TIR18 samples "with" and "without" clear ice layers respectively. This is an order of magnitude lower than the seasonal variation interval (~2 ‰ for $\delta^{18}O$) and is thus not affecting the identification of annual layers. Given this, we consider that the effect of these "melt layers" is also probably negligible on the ions used for the dating.

**2.3 Dating technique**

Dating of the ice cores was performed by annual layer counting, using the seasonality of the water stable isotopes, and the seasonality of the smoothed ECM and of specific ions (mainly $nssSO_4^{2-}$, $SO_4^{2-}/Na^+$ ratio and MSA) when the signal of water stable isotopes was unclear. These chemical species were chosen as their concentrations are known to show seasonal variations: sodium concentrations peak in austral winter (Wagenbach et al., 1998) when the sea ice surface is larger (Thomas et al., 2019), MSA concentrations usually peak in austral summer when the biological activity is high but are subject to migration within the annual layer (Curran et al., 2002) and sulfate concentrations also peak in austral summer, even though biological activity is not the only source of sulfate as it also comes from sea salt and other limited sources such as volcanism (intermittent high signals), terrestrial dust and anthropogenic activities (Wolff et al., 2006).

The identification of annual layers is sometimes challenging in coastal ice cores. The complete dating procedure we used is detailed in Appendix D and described briefly here. We used Matchmaker, a MATLAB® application, to visualize our multiple records and identify the annual layers (Rasmussen et al., 2008). After the first annual layer counting on the entire lengths' records, the relative dating obtained is refined by slightly adjusting the date attributed to some peaks larger than 3σ in the normalized $nssSO_4^{2-}$ and ECM records, as these peaks are considered as potential absolute age markers of volcanic eruptions

(Appendix E). The resulting dating has been used to train StratiCounter, the automated layer counting algorithms (Winstrup et al., 2012). Because of the high noise, the interannual variability of the annual layers and the presence of trends in our records, the conditions are not optimal to run the algorithms that require batches of approximately 50 layers sharing common characteristics (Winstrup et al., 2016). When using StratiCounter with five of the tiepoints identified as volcanic eruptions in our above-mentioned dating procedure (Pinatubo - 1991, Agung - 1963, Cerro Azul - 1932, Makian - 1861, Tambora - 1815;

Sigl et al., 2013), the coherence between our manual dating and the automatic dating and its narrow range of uncertainties supports the use of our manual dating to determine SMB.

### 2.4 Derived data

### 2.4.1 Surface mass balance records

Combining raw annual layer thicknesses with density profile allows to determine the annual layer thicknesses expressed in

meter ice equivalent (m i.e.). These are then corrected for ice flow to consider the lateral deformation of the ice due to the weight of the snow/firn column. The deepest and hence oldest part of the record is the most affected by the ice flow and oldest SMB is thus underestimated if the ice flow deformation is not accounted for.

The vertical strain rate correction is obtained from Phase-sensitive radar (ApRES) measurements. Following the methodology

outlined in Kingslake et al. (2014), a precise radar measurement in the vicinity of the ice core was carried out during two consecutive field seasons, making the measurements one year apart. Radar antennas were placed in exactly the same position each time, with receiver and transmitter antenna spaced 5 m apart. Vertical displacements of englacial reflectors in both measurements were translated into vertical velocity after correction for density changes based on the density profiles obtained from both ice cores. Surface vertical velocity is obtained by considering ice frozen to the bed and a zero vertical velocity at

the ice/bed interface. Errors on the velocity profile increase with depth and maximum errors are of the order of 0.02 m a$^{-1}$ near the ice/bed interface and <0.01 m a$^{-1}$ at the bottom of the ice cores. We then apply Eq. (15.3) from Cuffey and Paterson (2010) to correct strain thinning using the measured vertical velocity profile. Strain-rate corrections are up to 30-40 % for the deeper layers of the ice core and slightly higher compared to simple strain-rate correction methods based on a constant vertical strain (Nye method) or linear strain changes as used in Philippe et al. (2016).


Surface mass balance reconstructed from ice cores can be associated with large uncertainties. The typical sources of uncertainties are the analytical uncertainty of the depth-density profile (here <3 %) and uncertainties in the age-depth profile. Rupper et al., (2015) consider two types of uncertainties in the age-depth profile: the "peak-identification uncertainty" – which is the error in estimate of the number of peaks in a given section of the core – and the "peak-date uncertainty" – which is the

error in estimate of the absolute date of a given depth and comes from the assumption that the distance between two adjacent peaks is exactly one year. The "peak-date uncertainty" in our records is linked to the sampling resolution (5 cm) which limits the precision of the location of summer peaks. The annual layer thicknesses are mostly much larger than this resolution, which suggests that the probability is low to miss annual layers. The use of multiple seasonal parameters (here isotopes, ions and ECM) and absolute age markers (volcanic horizons) is known to considerably improve the peak-identification. We therefore

associate the "peak-identification uncertainty" to the discrepancy in the total number of peaks identified by the various operators in the manual dating process (<5 years). Section 3.2 details the quantification of SMB uncertainties in our cores.

### 2.4.2 Investigation of trends and seasonality

To study the potential trends affecting the main species ($\delta^{18}$O, d-excess, nssSO$_4^{2-}$, SO$_4^{2-}$/Na$^+$ and MSA), their annual mean is calculated by averaging the data within each annual interval (which are defined by the layer counting). An 11-year running

mean is also chosen and applied to the annual means to smooth the interannual variability and high frequency climate variability without damping the potential trends and cycles on longer time scales.

To study the seasonality of the main species, the measured data within each annual interval are interpolated over a 12-months period and then averaged for each month over the entire record's period. This regular interpolation over 12 months is based on the hypothesis that snow accumulation is constant during the year. The monthly climatology of RACMO2.3 between 1979 and 2016 at the two ice core sites (Lenaerts et al., 2017) and a complete year of snow height changes from an AWS located on Lokeryggen ice rise do not show any particular pattern of accumulation and thus both support the validity of this hypothesis (see Appendix Fig. E1).

## 2.5 Data validation

As also recognized by Thomas et al. (2022), measurements for water isotopes and chemical proxies are only available from ice cores. To our knowledge, there is no alternative sources for these datasets to be directly compared to. However, as discussed in section 4.1, preliminary age models for the top part of our cores have recently been used to discuss the representativeness of local SMB obtained from ice core data for estimates of regional surface mass balance derived from ground-penetrating radar profiles. Large-scale remote sensing estimates of Antarctic mass balance usually use the outputs of regional atmospheric models such as RACMO for surface mass balance estimates (e.g. Rignot et al., 2019) which cannot per se be used as a validation for direct measurements. In section 4, we compare our datasets to those from other cores in the Dronning Maud Land region, confirming the strong regional variability both in terms of absolute values and trends, which hinders a thorough validation process.

## 3 Results

### 3.1 Age models

The identification of annual layers was sometimes challenging, because of the high noise and background levels due to the coastal location of the ice cores, but the refinement of the age model with the identification of volcanic horizons reduces the uncertainties resulting from the manual annual layers counting. With 7 and 9 volcanic eruptions identified in FK17 and TIR18 respectively, the attribution of volcanic horizons to the ECM records (Appendix Fig. F1) was less successful than to the nssSO$_4^{2-}$ records with 9 and 11 volcanic eruptions identified in FK17 and TIR18, respectively (Fig. 2). The commonly used threshold of 2σ to consider peaks as potential volcanic horizons (e.g. Kaczmarska et al., 2004; Philippe et al., 2016) would strongly increase the number of attributions but it would also enhance the number of unattributed peaks since the ECM records from coastal ice cores are known to be characterized by high background signals due to the proximity of the ocean. The volcanic eruptions identified in both FK17 and TIR18 records are Ulawun - 2000, Pinatubo - 1991, El Chichon - 1982, Fuego - 1974, Agung - 1963, Cerro Azul - 1932, Tarawera - 1886, Makian - 1861 and Tambora - 1815. The additional eruptions identified in TIR18 are Santa Maria - 1902, Krakatoa - 1883, Cosiguina - 1835 and Galunggung - 1822. The Ulawun eruption (2000) was recently identified in the coastal DML region, to the west of our cores sites (Ejaz et al., 2021).

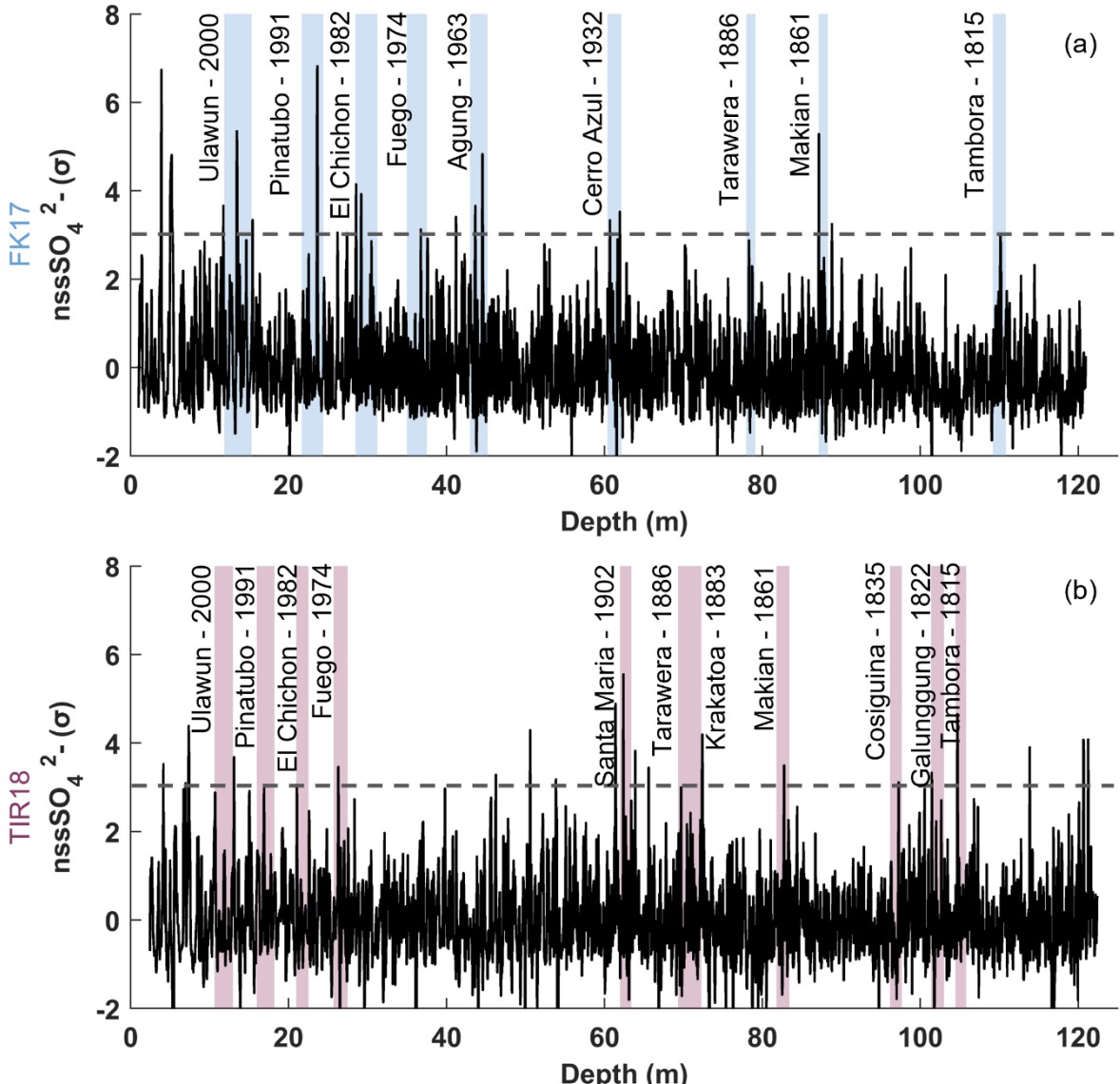

**Figure 2. Normalized nssSO$_4^{2-}$ records for FK17 (a) and TIR18 (b). The signal (black line) is expressed as a multiple of standard deviation (σ), the 3σ threshold is the dotted horizontal line, and identified volcanic peaks are shown as blue- or burgundy-colored bars, for FK17 and TIR18 respectively. The thicknesses of the color bars are related to the extended period during which the volcanic signal is potentially recorded in the ice core (year of the eruption + 2 years).**

The age-depth profiles of FK17 and TIR18 are presented in Fig. 3 together with the automatic dating from StratiCounter and its range of uncertainties. These age models obtained by manual dating (colored lines in Fig. 3) differ from the automatic ones (black lines in Fig. 3) by 0 to 3 years for FK17 and by 0 to 5 years for TIR18 and they mostly lie within the uncertainties of the automatic age models (grey shadings in Fig. 3), except for short sections at maximum ± 3 years from the narrow range of uncertainties. A total of 225 annual layers (226 in StratiCounter) have been identified in the FK17 records and 239 annual

layers (241 in StratiCounter) in the TIR18 records, the ice cores are hence dated back to CE 1793 ± 3 years for FK17 and 1780 ± 5 years for TIR18.

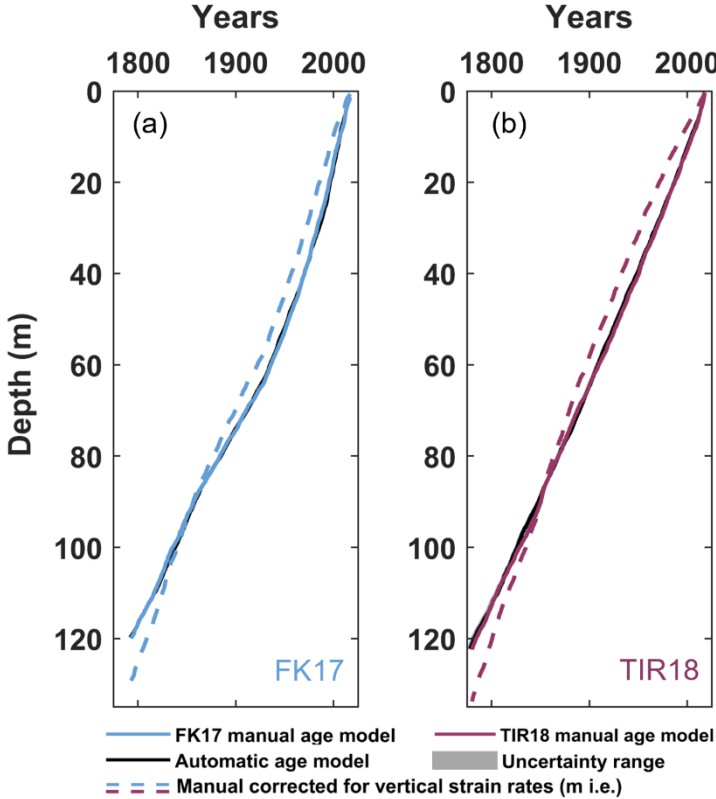

**Figure 3. Age–depth profile reconstructed from the manual layer counting process for (a) FK17 (blue line) and (b) TIR18 (burgundy line). Black lines and grey shading are hardly distinguishable, they respectively show the automatic dating and the uncertainty range derived from StratiCounter. The dotted lines represent the manual age models expressed in m i.e. and corrected for vertical strain rates.**

Although they reach similar dates at 120 m, as a result of near identical mean SMBs over the whole period (see below, Sect. 3.2.), the two uncorrected age models (solid color lines in Fig. 3) differ in shape, with a sub-linear profile for TIR18 and a more curved and undulatory profile for FK17. The correction of the two age models for vertical strain rates (dotted color lines in Fig. 3) brings FK17 closer to linear, while TIR18 now develops a concave curvature after 1850 (above 100 m). These peculiarities are related to those of the surface mass balance records described in the next section.

## 3.2 Surface mass balance records

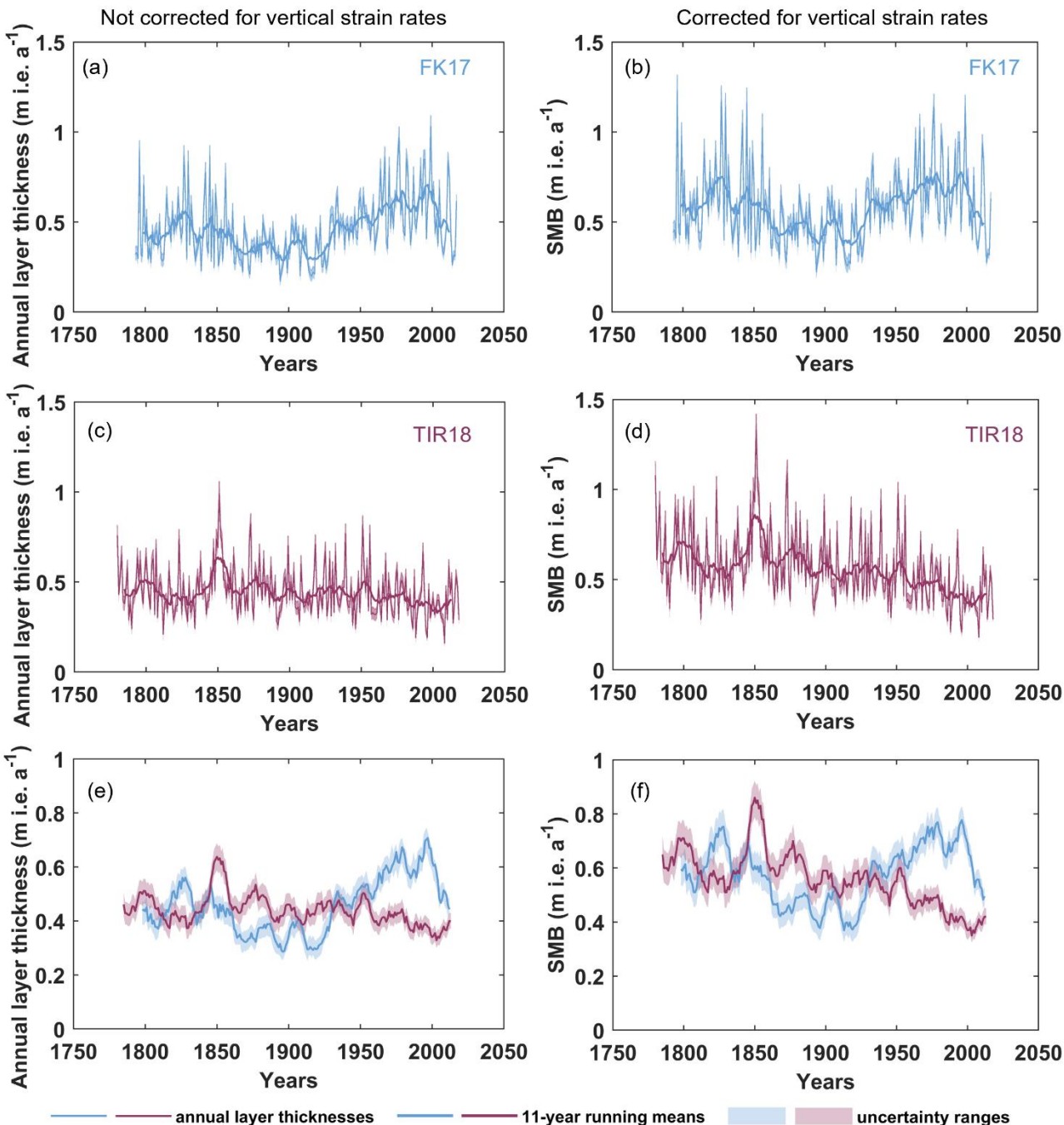

**Figure 4.** Annual layer thickness (left panels) and surface mass balance (right panels) in meter ice equivalent per year (m i.e. a⁻¹) (a-b) for FK17 core, (c-d) for TIR18 core. The thin lines connect the annual layer thicknesses and the thick lines are the 11-year running means. The shaded colored ranges show the uncertainties associated to the SMB. The left panels represent the annual layer thicknesses not corrected for strain rates and the right panels represent the annual layer thicknesses corrected for strain rates. (e-f) The smoothed SMB records result from the use of an 11-year running mean on SMB.

The SMBs at the two ice core locations are presented in Fig. 4. As described in section 2.4.1, the SMB is associated with uncertainties. Here, the analytical uncertainty of the depth-density profile is inferior to 3 %, "the peak-date uncertainty" is linked to the sampling resolution (5 cm) and the "peak-identification uncertainty" is estimated based on the discrepancy between operators' manual relative datings, i.e. maximum 5 years (see Appendix D for the detailed dating procedure). The error on the vertical strain rates (< 0.01 m a⁻¹) must be added to define the uncertainties of SMB corrected for vertical strain rates. We thus define the SMB uncertainties using error propagation calculation which results in the shaded colored ranges in Fig. 4. For both FK17 and TIR18 records, the average uncertainty for the whole period is ± 0.04 m i.e. a⁻¹ for the annual layer thickness (without correction) and ± 0.05 m i.e. a⁻¹ for SMB with vertical strain rates correction. The mean annual layer

thickness (without vertical strain rates correction) is $0.46 \pm 0.04$ m i.e. $a^{-1}$ for FK17 and $0.44 \pm 0.04$ m i.e. $a^{-1}$ for TIR18 (Fig. 4a and Fig. 4c, respectively). These are compared to the SMB corrected for vertical strain rates in Fig. 4b and Fig. 4d for FK17

(mean: $0.57 \pm 0.05$ m i.e. $a^{-1}$) and TIR18 (mean: $0.56 \pm 0.05$ m i.e. $a^{-1}$) respectively. This comparison highlights the crucial need to correct the annual layer thicknesses for vertical strain rates when analyzing potential long-term trends in ice core records. For both FK17 and TIR18, two 11-year running means have been calculated (Fig. 4e and Fig. 4f): one on the annual layer thickness not corrected for vertical strain rates (left panels) and one on the SMB corrected for vertical strain rates (right panels). These running means seem to indicate the presence of some long-term cyclicity in all SMB records. However, this

might be a "side effect" of the signal treatment and its pertinence should be confirmed in the future by the use of time-series analysis tools.

All records exhibit large interannual variability (Fig. 4b and Fig. 4d). The mean SMB is similar for FK17 and TIR18 but the annual records are very different. FK17 shows a long-term oscillating behavior with an increasing trend between 1793 and

355 ~1825, followed by a decrease until ~1925, then a new increase and a plateau until ~1995, and finally a recent decreasing trend. TIR18 shows higher variability and no detectable trend in the oldest part of the record (before 1850), followed by a significant decreasing trend until present day. These trends have been identified using the ensemble algorithm BEAST ("Bayesian Estimator of Abrupt change, Seasonal change, and Trend"; Zhao et al., 2019).

We defined four different time periods (~200, 100, 50 years and the last 20 years) to evaluate SMB changes or trends and their relative intensity as one moves to more recent times (Table 2). All time periods start in 1816 because of the well-defined Tambora marker.

| Period (years CE) | SMB (m i.e. $a^{-1}$) | |
| --- | --- | --- |
| | FK17 | TIR18 |
| 1816-2017 | $0.57 \pm 0.05$ | $0.55 \pm 0.05$ |
| 1816-1917 | $0.53 \pm 0.05$ | $0.61 \pm 0.06$ |
| 1918-2017 | $0.62 \pm 0.05$ | $0.48 \pm 0.04$ |
| | +16 % | -20 % |
| 1816-1967 | $0.55 \pm 0.05$ | $0.58 \pm 0.05$ |
| 1968-2017 | $0.65 \pm 0.05$ | $0.43 \pm 0.04$ |
| | +18 % | -27 % |
| 1816-1997 | $0.57 \pm 0.05$ | $0.56 \pm 0.05$ |
| 1998-2017 | $0.57 \pm 0.04$ | $0.40 \pm 0.04$ |
| | 0 % | -28 % |

**Table 2. Mean SMB at FK17 and TIR18 for different time periods. These SMB are corrected for vertical strain rates. The % refers, in each case, to the change between the two compared time windows.**

As for the whole time period, FK17 and TIR18 mean SMBs are quite similar between 1816 and 2017. This however hides strong differences in the SMB dynamics. For the 1918-2017 period, the mean SMB is $0.62 \pm 0.05$ m i.e. $a^{-1}$ for FK17 and $0.48 \pm 0.04$ m i.e. $a^{-1}$ for TIR18. This represents a 16 % increase and a 27 % decrease, for FK17 and TIR18 respectively, compared to the previous period (1816-1917). For the 1968-2017 period, the mean SMB is $0.65 \pm 0.05$ m i.e. $a^{-1}$ for FK17 and $0.43 \pm 0.04$ m i.e. $a^{-1}$ for TIR18, representing an 18 % increase for FK17 and a 27 % decrease for TIR18 compared to the 1816-1967

period. And for the 1998-2017 period, the mean SMB is $0.57 \pm 0.04$ m i.e. $a^{-1}$ for FK17 and $0.40 \pm 0.04$ m i.e. $a^{-1}$ for TIR18, which corresponds to a status quo for FK17 and a decrease of 28 % for TIR18 compared to the previous period (1816-1997). This highlights the contrast between the continuous and increasingly pronounced decrease of TIR18 SMB and the on average

increasing SMB at FK17 over the last century, except for the last 20 years marked by decreasing SMB values from 1995 onwards.

## 3.3 Seasonal cyclicity, pluriannual cyclicity and trends

The left and central panels of Fig. 5 show that, in both records, $\delta^{18}O$, nssSO$_4^{2-}$ and SO$_4^{2-}$/Na$^+$ peak in austral summer and so does the d-excess but its seasonality is less marked. MSA peaks in austral winter in FK17 and in late autumn in TIR18, while it usually peaks in austral summer when the biological activity is high (Curran et al., 2002). The previous observations are valid for both the medians and the 0.25-0.75 quartiles. The SO$_4^{2-}$/Na$^+$ ratios as well as MSA and nssSO$_4^{2-}$ concentrations are significantly lower in the TIR18 records than in the FK17 records. The $\delta^{18}O$ signal is more negative in TIR18 than in FK17 while d-excess is slightly higher in TIR18 than in FK17, although at the limit of uncertainties (see Table 3).

The right panel of Fig. 5 displays the pluriannual cyclicities and trends based on an 11-year running mean of the annual averages for the $\delta^{18}O$, d-excess, MSA, nssSO$_4^{2-}$ and SO$_4^{2-}$/Na$^+$ records of FK17 and TIR18. In terms of spatial variability, the relative level of FK17 and TIR18 proxies highlighted in the seasonality panels is confirmed in this smoothed long-term record.

In terms of temporal variability, Table 3 summarizes mean values for two time periods (1816-1950 and 1951-2015) to seek for different trends before and after the recent enhanced anthropogenic impacts. The means for the entire profile records ([1793-2017] for FK17 and [1780-2018] for TIR18) are also presented in Table 3.

| | $\delta^{18}O$ | | d-excess | | MSA | | nssSO$_4^{2-}$ | | SO$_4^{2-}$/Na$^+$ | |
|---|---|---|---|---|---|---|---|---|---|---|
| | FK17 | TIR18 | FK17 | TIR18 | FK17 | TIR18 | FK17 | TIR18 | FK17 | TIR18 |
| 1816-1950 | -19.14 | -19.55 | 5.70 | 6.00 | 43.5 | 31.5 | 126.7 | 62.8 | 1.0 | 0.7 |
| 1951-2015 | -18.93 | -19.25 | 5.13 | 5.61 | 36.1 | 22.4 | 149.5 | 68.3 | 1.3 | 0.8 |
| mean | -19.12 | -19.46 | 5.63 | 6.00 | 41.1 | 29.3 | 130.4 | 64.2 | 1.1 | 0.7 |

**Table 3. Mean values of $\delta^{18}O$, d-excess, MSA, nssSO$_4^{2-}$ and SO$_4^{2-}$/Na$^+$ of FK17 and TIR18 for two time periods (1816-1950 and 1951-2015) and for the entire records ([1793-2017] for FK17 and [1780-2018] for TIR18).**

For both records, the $\delta^{18}O$ values are less negative, d-excess and MSA are lower, and nssSO$_4^{2-}$ and SO$_4^{2-}$/Na$^+$ are higher during the 1951-2015 period than in the previous 1816-1950 period. This represents a 1-2 % change in $\delta^{18}O$ values, a decrease of 10-6 % for d-excess, a 17-29 % decrease for MSA concentrations, a 18-9 % increase in nssSO$_4^{2-}$ and an increase of 30-18 % for the SO$_4^{2-}$/Na$^+$ ratio, in FK17-TIR18 records respectively. All records show large interannual variability and potential cycles with different and variable periods. Increasing trends in nssSO$_4^{2-}$ and SO$_4^{2-}$/Na$^+$ appear more clearly in the FK17 record than in the TIR18 record.

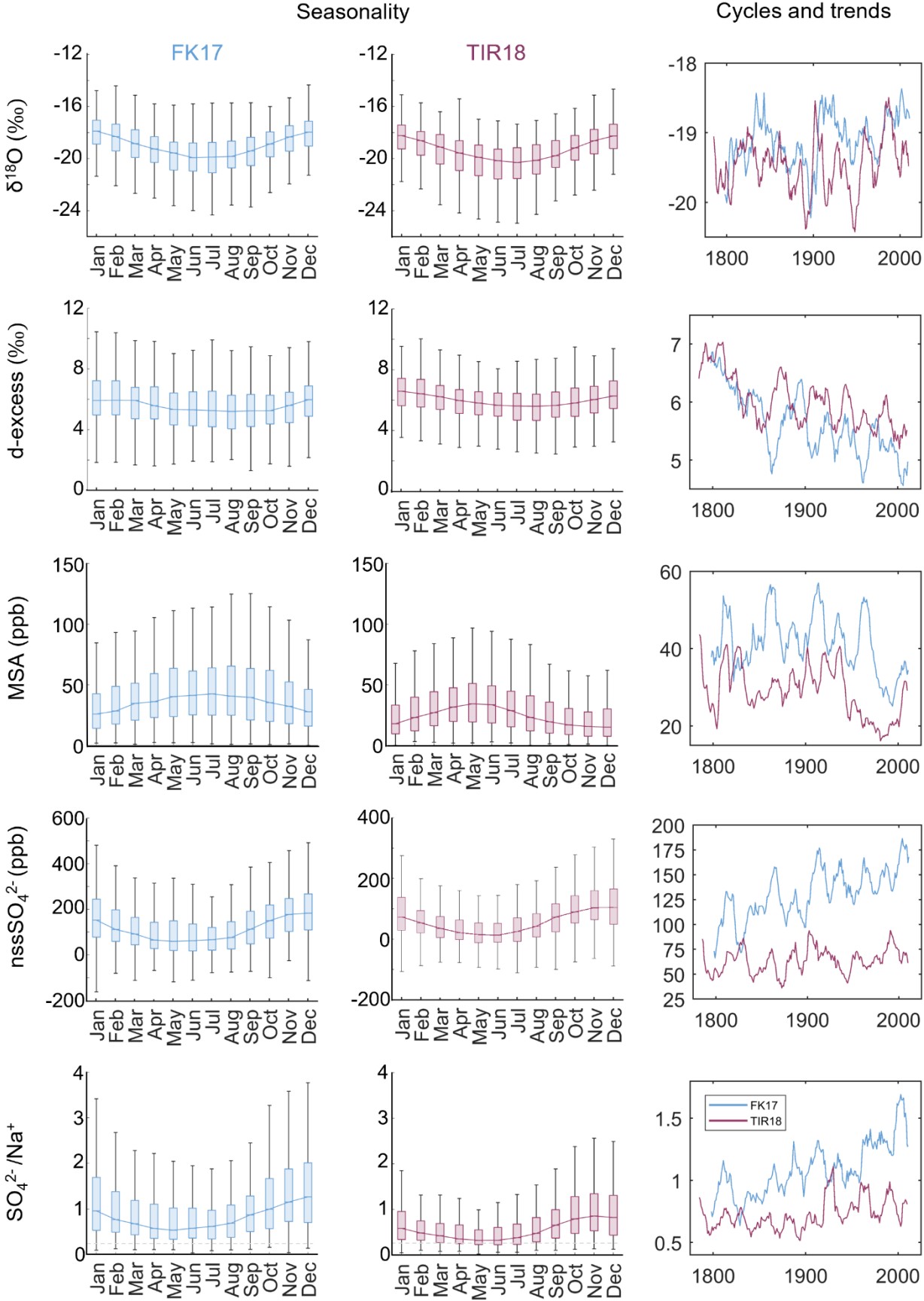

**Figure 5. Seasonality and trends analyses of the main species used for dating over the whole dataset (from top to bottom: δ¹⁸O, d-excess, MSA, nssSO₄²⁻ and SO₄²⁻/Na⁺). The left and central panels (FK17 and TIR18 respectively) show the seasonality with the monthly medians. The boxes represent the range of values between the 0.25 and 0.75 quartiles and the line connects the medians. The vertical lines are the extreme values of the range. Note that for visualization, the outliers are not represented here. Also note the different concentration range used for nssSO₄²⁻. The horizontal line in the SO₄²⁻/Na⁺ panels is the reference weight sea water ratio**

of 0.25. The right panel shows the annual means signal that has been smoothed with an 11-yr running mean, for FK17 in blue and TIR18 in burgundy.

## 4 Discussion and perspectives

We present here two information-rich datasets both in terms of direct environmental proxies, with the continuous measurements of isotopes and chemical species, and derived information such as surface mass balance records and long-term trends. Both environmental proxies and derived data show contrasting behaviors, suggesting strong spatial and temporal variability at regional scale.

### 4.1 Regional representativeness of ice cores

Kausch et al., (2020) showed that there is an important snowfall-driven contrast between the windward and the leeward sides of the ice rise, with a SMB up to 1.5 times higher on the windward side of the Lokeryggen ice rise (LIR) between 1983 and 2015. They also showed the presence of a local SMB minimum due to wind erosion at the peak of the ice rise where the ice core is drilled and suggest that the SMBs derived from ice cores would rather be representative of the surrounding ice shelf. Cavitte et al., (2022) showed that, at LIR and HIR, the ice cores SMBs are representative of a small surface area of the ice rise, typically ~200-500 m radius around the drilling site and are systematically 0.08-0.16 m i.e. $a^{-1}$ lower than the mean SMB value calculated for the whole ice rise. They conclude that ice cores are sufficient to obtain an accurate estimation of the multi-annual to decadal variability of SMB at the regional scale and that the SMB records should be adjusted (e.g., using ground-penetrating radar data) to be more representative of the entire ice rise region if the aim is to study the SMB at the regional scale or to compare it to RCM simulations.

More than providing age models to other methods, our datasets also present a large number of development possibilities for the interpretation of the various paleo profiles which will be discussed at length in companion papers. These are, however, briefly outlined in the following sections.

### 4.2 Long-term trends and spatial variability in the paleo proxy records

Many species measured in FK17 and TIR18 records can be used as paleo proxies. Our proxy records seem to present some potential long-term cyclicity and trends, these are discussed here and compared to the "IC12" record of the Derwael ice rise (DIR) located ca. 100 km to the east of LIR (Philippe et al., 2016; Philippe and Tison, 2023; and Appendix G – Fig. G1 and G2 and Table G1). The isotopic composition of the ice ($\delta^{18}O$ and $\delta D$) is related to the temperature gradient between the evaporation site and the condensation site and has thus been widely used to reconstruct past changes in temperature (e.g. Dansgaard, 1964; Jouzel et al., 1987). Stable water isotopes have also been proposed as proxy to reconstruct past sea ice variability during the recent centuries (Ejaz et al., 2021). The $\delta^{18}O$ values are less negative during the 1951-2015 period than in the previous 1816-1950 period for the three records, which suggests higher temperature during the precipitation events (Appendix Fig. G1a). The mean $\delta^{18}O$ value of IC12 (-18.47 ± 0.05 ‰) is significantly higher than the means of FK17 and TIR18 (-19.12 ± 0.02 ‰ and -19.46 ± 0.02 ‰, respectively), implying higher temperature at DIR than at LIR and HIR, although it has been shown that several other factors might influence the $\delta^{18}O$ records, such as conditions at the source, transport, precipitation, sea ice extent and post-depositional processes like diffusion (Naik et al., 2010; Ejaz et al., 2021; Sinclair et al., 2012). Our records could thus be used to produce a past temperature reconstruction as done by Ejaz et al., (2022), using the $\delta^{18}O$ record from an ice core and ERA5 surface air temperature and then investigating the potential role of the climate modes such as El Niño Southern Oscillation, the Interdecadal Pacific Oscillation and the Southern Annular Mode in the temperature variability.

Deuterium excess (d-excess), which is derived from $\delta^{18}O$ and $\delta D$, is a tracer of precipitation origin and a proxy of the temperature and relative humidity at the evaporation site (Stenni et al., 2010). Both FK17 and TIR18 d-excess records show a long-term decreasing trend (Fig. 5 and Table 3), suggesting an increasingly lower temperature at the evaporation site for both ice cores. This is contrasting with the d-excess record of IC12 showing no trend (Appendix Fig. G1b), which is confirmed in Appendix Table G1. This IC12 record of d-excess is also significantly higher than FK17 and TIR18 records, which might indicate an evaporation site with higher temperature for IC12. Taken together, these assumptions tend to suggest a different precipitation source for the IC12 site. All three records display some long-term cyclicities (Appendix Fig. G1a and b) that are worth being further investigated, using e.g. statistical approaches such as Multivariate Singular Spectrum Analysis (MSSA) or Multitaper Method (MTM).

The MSA, nssSO$_4^{2-}$ and weight SO$_4^{2-}$/Na$^+$ records are unfortunately not complete for IC12. The 11-year running means of the continuous top part and some deeper intervals are presented in Appendix Fig. G2 and in Table G1. The IC12 mean concentration of MSA is lower than in FK17 and slightly lower than in TIR18 (Appendix Table G1), the nssSO$_4^{2-}$ values of IC12 are close to TIR18 concentrations before 1950 and in between TIR18 and FK17 concentrations after 1950 (Appendix Fig. G2b). The top part of the SO$_4^{2-}$/Na$^+$ IC12 record is remarkably similar to the TIR18 record.

Non-sea-salt sulfates (nssSO$_4^{2-}$) have different sources but they are mainly produced by biological activity (Wolff et al., 2006) and can hence be used as a proxy for past sea ice extent. The nssSO$_4^{2-}$ concentrations of FK17 are twice the ones of TIR18 and display an increasing trend on the long-term (Fig. 5), which is not the case in TIR18 where there is only a smaller increase (9 %) between the 1816-1950 period and the 1951-2015 period (Table 3). Both FK17 and TIR18 show a multidecadal pseudo-cyclicity. The SO$_4^{2-}$/Na$^+$ ratio gives information on the plausible depletion of SO$_4^{2-}$ with respect to seawater composition and could thus be used as an indicator of interactions with sea ice. Indeed, when sea ice forms and cools down, the salinity of the brines increases, resulting in precipitation of saline compounds such as mirabilite (Na$_2$SO$_4 \times$ 10 H$_2$O) at temperatures below -8 °C, leading to a depletion of the SO$_4^{2-}$/Na$^+$ ratio in the remaining brines relative to bulk sea water, with a weight ratio of 0.25 for SO$_4^{2-}$/Na$^+$ in sea water (Alvarez-Aviles et al., 2008; Abram et al., 2013). As for the nssSO$_4^{2-}$ concentrations, the SO$_4^{2-}$/Na$^+$ weight ratios show a general increasing trend in the FK17 record, while it is less clear in the TIR18 record which exhibits rather contrasting values before and after ~1915 (Fig. 5). Both SO$_4^{2-}$/Na$^+$ records are always higher than the sea water weight ratio (0.25), indicating a dominant impact of the nssSO$_4^{2-}$ on the SO$_4^{2-}$ signature. The calculated nssSO$_4^{2-}$ values using the classical sea water ratio as a reference represent 44 to 66 % of the total SO$_4^{2-}$ records, for TIR18 and FK17 respectively. However, these could be minimal values, since ssSO$_4^{2-}$ fractionation could have occurred but been partly masked by the nssSO$_4^{2-}$ contribution to the total SO$_4^{2-}$ record (Vega et al., 2018). Negative nssSO$_4^{2-}$ values observed in both records (see data base) confirm this hypothesis. This thus indicates a SO$_4^{2-}$ depletion and a source of sea salt material that is fractionated (Abram et al., 2013), which has already been observed in other coastal ice cores (Abram et al., 2013 and reference therein; Vega et al., 2018). For most of the last decade, frost flowers have been considered the most probable source of fractionated sea salt aerosol reaching coastal and inner Antarctica (Rankin et al., 2002). However, some recent studies (Yang et al., 2008; Huang and Jaeglé, 2017; Vega et al., 2018) point to an alternative mechanism involving the sublimation of blowing salty snow. An extensive study of SO$_4^{2-}$/Na$^+$ and nssSO$_4^{2-}$ (such as in Vega et al., 2018) would allow to better understand the mechanisms responsible for sea-salt aerosol production (including fractionation), transport and deposition at our coastal sites and potentially explain the observed contrast between them at the regional scale. MSA is produced by biological activity in the sea ice zone and many correlations between MSA and sea ice extent have been observed at coastal sites in Antarctica, allowing the reconstructions of past sea ice extent (Thomas et al., 2019 and references therein). The MSA concentrations in both FK17 and TIR18 records are characterized by a large decrease during the 1951-2015 period compared to the previous 1816-1950 period (Table 3). During storage of the ice cores, MSA is able to diffuse through solid ice which might result in MSA loss (Abram et al., 2008).

The shallower parts of our cores were, however, analyzed within 30 months after drilling and the loss is thus expected to be low. Several processes could be invoked to explain the observed recent MSA decrease in our ice cores: reduced biological production, less efficient transport of MSA towards the ice core location, increased post-depositional MSA losses due to changing environmental conditions… Note that the increasing $nssSO_4^{2-}$ concentrations, particularly in FK17, suggest increasing biological activity, which might rule out the first option

On the spatial contrast between FK17 and TIR18, the more negative $\delta^{18}O$ signal and the slightly higher d-excess in TIR18 than in FK17 (Table 3) might indicate a more "continental" influence on the precipitation at HIR than at LIR, with colder temperature and a more distant origin of evaporation. This hypothesis is supported by significantly lower $SO_4^{2-}/Na^+$ ratios and MSA and $nssSO_4^{2-}$ concentrations in the TIR18 records than in the FK17 records, pointing to lower impurities input and less influence of the sea ice.

### 4.3 Seasonality of the proxies

Given the high accumulation at our ice cores sites, we were able to interpret our records at a monthly resolution, allowing to study their seasonality. For example, in our records, after a few years with summer peaks (not shown), MSA peaks in austral winter in FK17 and in late autumn in TIR18 (Fig. 5). Since it usually peaks in austral summer when the biological activity is high (Curran et al., 2002), this indicates that MSA is subject to post-depositional movement within the annual layers. This phenomenon has been observed in several records (e.g. Curran et al., 2002; Thomas and Abram, 2016; Hoffmann et al., 2022) and the exact mechanism is not understood yet. Curran et al., (2002) suggest a complex interaction between the gradients in $nssSO_4^{2-}$, $NO_3^-$ and sea salts (which are influenced by the accumulation). In a more recent study, Osman et al., (2017) report that the depth at which the MSA movement begins varies with the mean annual accumulation rate, particularly in Antarctic coastal regions, and they identify a critical density value of $\sim550$ kg m$^{-3}$ for movement to begin. $Na^+$ ion concentration is also pointed out as of major influence on MSA migration (Osman et al., 2017). Our records represent a new opportunity to verify these hypotheses further and to better understand the underlying processes.

Looking at the seasonality of the $SO_4^{2-}/Na^+$ ratios, a maximum in summer and a minimum in winter are expected since the sulfate is known to peak in summer (when the biological activity peaks) and the sodium peaks in winter (with the source being the sea ice), this is what has been observed in both FK17 and TIR18 records (Fig. 5). It is worth noting that the median ratio is always higher than the bulk sea water ratio for both records, with lower values in April-May-June (0.32 for TIR18 – closer to the sea water weight ratio – and 0.54 for FK17), indicating a dominant contribution of $nssSO_4^{2-}$ to the signal throughout the year, potentially masking some contribution from $ssSO_4^{2-}$ fractionation during winter.

Our records thus represent an additional asset to study the seasonality of aerosols in coastal ice cores. This also raises a great opportunity to better understand the link between the atmospheric processes and the signal preserved in ice cores. The use of air mass back-trajectories is an option to study this link and might shed light on the observed local contrasts between the different sites.

### 4.4 Surface mass balance records

Better understanding processes for individual proxies might also help to understand the more complex SMB records also showing strong spatial and temporal variability (Fig. 4). Given this variability, it is crucial to work on identical time windows when comparing several ice cores, which is the case of this study (see Appendix G). The strong spatial and temporal variability contrasts are even more remarkable when considering the IC12 SMB record (Appendix Fig. G1c) which is also different, with stable (but variable at the interdecadal scale) SMB between 1750 and 1950 followed by a strong increase until the end of the

record in 2011. Both FK17 and TIR18 SMBs records are higher than the IC12 record before 1900. After 1900, the TIR18 SMB record is relatively similar to IC12 until 1950 when IC12 shows a strong increase (+ 19 cm between the 1816-1961 period and the 1962-2011 period, see Appendix Table G2) and TIR18 shows a strong decrease (- 16 cm between the 1816-1961 period and the 1962-2011 period, see Table G2). The mean SMBs calculated over the "last" 100, 50 and 20 years for IC12 and FK17 (Appendix Table G2) are close, even though the SMB of FK17 starts to decrease from 1995 onwards while it keeps increasing in the IC12 record (Appendix Fig. G1c). It is however interesting to note, that despite these strong interdecadal differences between the three ice rises, their mean SMB for the whole period is very similar (Appendix Table G2).

Studying adjacent ice rises has been done recently in the area of the Fimbul ice shelf (DML), although the records cover shorter timescales (Vega et al., 2016). Three firn cores (~20 m) from the Kupol Ciolkovskogo, Kupol Moskovskij and Blåskimen Island ice rises and a longer core (100 m) from the ice shelf were drilled, allowing to study the spatial variability of the surface mass balance for the last 2 to 5 decades for the firn cores and for 250 years for the 100 m core. Two of the firn cores showed no significant long-term trend during the 2 decades of the records, and the third one showed a weak decreasing trend along its 50 years record, similar to the decrease observed in the 100 m core (Vega et al., 2016; Vega et al., 2018). Our two records combined with the earlier Derwael ice rise record in the same area are thus a great opportunity to further document the spatial variability observed in coastal DML, on longer timescales, and to look for the mechanisms (both depositional and post-depositional) explaining such contrasting results at the regional scale. These would also allow to compare the representativeness of the spatial variability reproduced (or not) in regional model outputs, an important target of the Mass2Ant project.

## 5 Data availability

FK17 and TIR18 datasets (https://doi.org/10.5281/zenodo.7848435) are merged in a file named "Physico-chemical properties of the top 120 m of two ice cores in Dronning Maud Land (East Antarctica)" (Wauthy et al., 2023) available on Zenodo under Creative Commons Attribution 4.0 International Public License. The data available are:

- Isotopes ($\delta^{18}O$, $\delta D$ and d-excess),
- Ions concentrations ($Na^+$, $K^+$, $Mg^+$, $Ca^+$, MSA, $Cl^-$, $SO_4^{2-}$ and $NO_3^-$),
- ECM,
- Density,
- Surface mass balance (SMB) not corrected and corrected for vertical strain rates.

The file is composed of a "Read me" sheet with general information: core location, instruments and their precision, units and resolutions of the records as well as notes and a summary on the errors. The two other sheets display the records for FK17 and for TIR18.

Additional datasets used for the figures:

- The IC12 annual layer thicknesses and age-depth are available on https://doi.org/10.1594/PANGAEA.857574 (Philippe et al., 2016).
- The top 103 meters (corresponding to 1815 and the Tambora eruption) of IC12 chemistry ($Na^+$, MSA, $Cl^-$, $SO_4^{2-}$ and $NO_3^-$) and water stable isotopes ($\delta^{18}O$, $\delta D$ and d-excess) are available on https://doi.org/10.5281/zenodo.7798252 (Philippe and Tison, 2023).

## 6 Code availability

- All analytical instruments used dedicated software provided with the equipment.
- Matchmaker, the graphical MATLAB® application used to identify annual layers, is presented in Rasmussen et al., (2008) and is available from S. O. Rasmussen upon request.

- StratiCounter, the automated layer counting algorithms (Winstrup et al., 2012), uses MATLAB® and is available on GitHub (https://github.com/maiwinstrup/StratiCounter).
- BEAST, the ensemble algorithm used to identify trends (Zhao et al., 2019), was run on MATLAB® and is available on GitHub (https://github.com/zhaokg/Rbeast).
- The figures were created using MATLAB® R2023a and basic plot functions.

## 7 Conclusion

Two ice cores were drilled at the crest of two adjacent ice rises (Lokeryggen and Hammarryggen) along the Princess Ragnhild Coast, East Antarctica. The ice cores are dated back to CE 1793 ± 3 years for FK17 and 1780 ± 5 years for TIR18 at a depth of 120 m, using water stable isotopes, ECM and mainly $nssSO_4^{2-}$, $SO_4^{2-}/Na^+$ ratios and MSA and the dating obtained is verified using StratiCounter, the automated layer counting algorithms. We define the annual surface mass balance corrected for vertical strain rates and calculate the annual means of the main species and their monthly seasonality.

The paleo proxy records present contrasting trends and long-term variability. Both FK17 and TIR18 records show a long-term decreasing trend in d-excess and $\delta^{18}O$ values are less negative during the 1951-2015 period than in the previous 1816-1950 period. These can be linked to changes in temperature at the evaporation site and during the precipitation events. The MSA concentrations in both FK17 and TIR18 records are characterized by a large decrease after 1950. The $nssSO_4^{2-}$ concentrations of FK17 are twice the ones of TIR18 and display an increasing trend on the long-term when there is only a small increase after 1950 in TIR18. The $SO_4^{2-}/Na^+$ ratios show a similar contrast between FK17 and TIR18 and are consistently higher than the sea water ratio, indicating a dominant impact of the $nssSO_4^{2-}$ on the $SO_4^{2-}$ signature (44 to 66 % of the total $SO_4^{2-}$, for TIR18 and FK17 respectively). However, this could be minimal values, since $SO_4^{2-}$ fractionation could occur but being partly masked by the $nssSO_4^{2-}$ (Vega et al., 2018).

The mean SMB is similar for FK17 and TIR18 (respectively 0.57 ± 0.05 and 0.56 ± 0.05 m i.e. $a^{-1}$) but the annual records are very different: starting from variable but relatively similar values for most of the 19th century, FK17 is globally increasing and TIR18 globally decreasing during the 20th century. In that respect, FK17 behaves similarly to the IC12 ice core collected at the summit of Derwael ice rise, located about 100 km east of LIR, despite a recent decrease in FK17 in the 21st century.

Further detailed study of the water stable isotopes, sea salt aerosols and other biogenic compounds presented here, linked e.g. to atmospheric pathways analysis such as air mass back-trajectories, will allow to better understand the mechanisms responsible for the production, transport, and deposition of the environmental proxies at our coastal sites. These records collected at high accumulation locations also provide a great opportunity to better understand the link between the atmospheric processes and the signal preserved in ice cores at the seasonal scale, and to decipher the reasons behind the regional contrasts in the records. In addition, confirming the relevance of multi-decadal cyclicity and understanding the long-term trends will shed more light on the complex interplay between anthropogenic and natural atmospheric forcings, since the latter might partly mask the first in the records. Finally, these datasets and their interpretation process-wise will further validate and potentially improve regional atmospheric models in Antarctica.

## 8 Appendices

*Appendix A – Automatic weather station records*

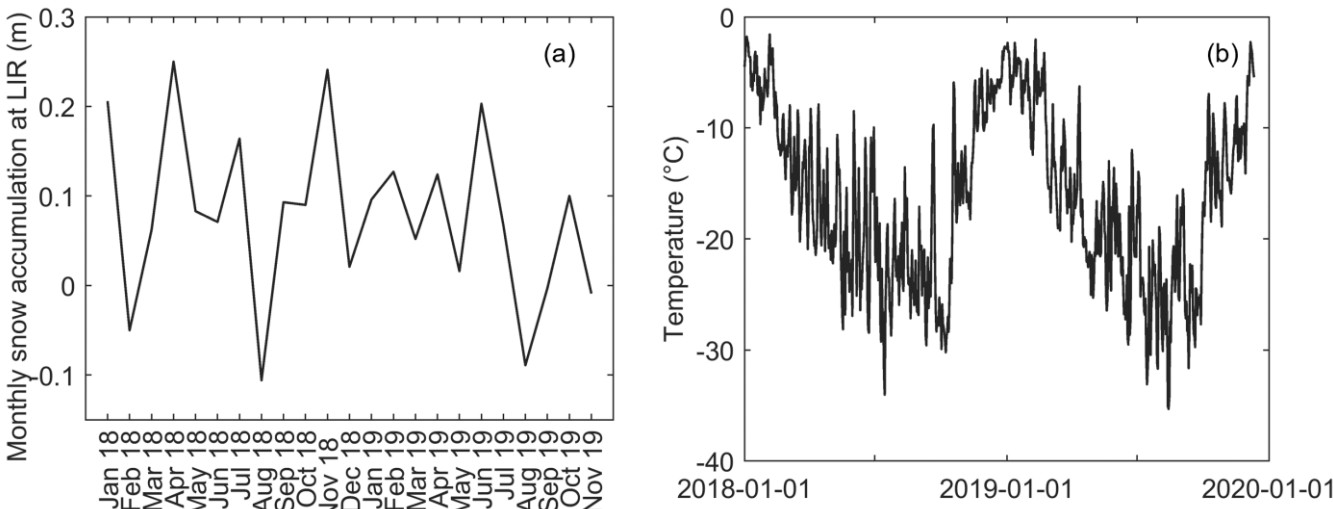

**Figure A1. Meteorologic records from the AWS on the eastern side of LIR. Two years (from the 1st of January 2018 to the 12th of December 2019) of monthly snow accumulation (a) and daily temperature (b).**

*Appendix B – Complements to the liquid chromatography method*

|  | Anions | Cations |
|---|---|---|
| Column | IonPac AS15 - Capillary | IonPac CS15 - Analytical |
| (dimensions) | (0.4 x 250 mm) | (3 x 250 mm) |
| Guard | IonPac AG15 - Capillary Guard | IonPac CG15 - Analytical Guard |
| (dimensions) | (0.4 x 50 mm) | (3 x 50 mm) |
| Temperature | 30 °C (IC Cube) | 40 °C (column) |
| Eluent | KOH - Isocratic 30 mM | MSA - Isocratic 30 mM |
| Eluent flow rate | 0.012 ml/min | 0.36 ml/min |
| Detector | Conductivity | Conductivity |
| Cell temperature | 35 °C | 35 °C |
| Suppressor | ACES 300 | CDRS 600 |
| Applied current | 9 mA | 32 mA |

**Table B1. Instrumental parameters for anions and cations measurement by liquid chromatography (Dionex-ICS5000).**

*Appendix C – Impact of melt layers on isotopic signals*

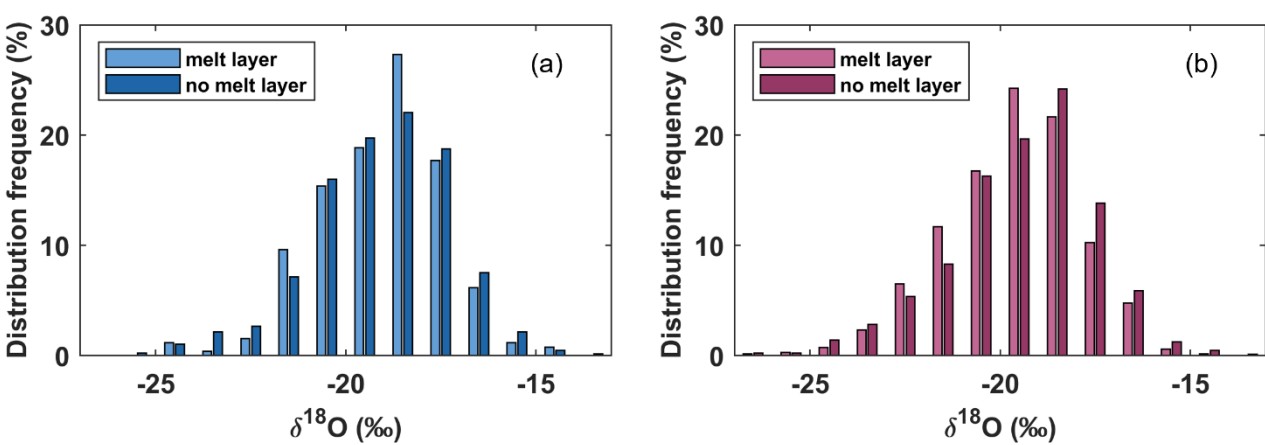

**Figure C1. Distribution frequency of the $\delta^{18}$O samples "with" (lighter color) and "without" (darker color) clear ice layers in FK17 (a) and TIR18 (b).**

The distribution frequencies of $\delta^{18}O$ samples "with" and "without" clear ice layers are shown for FK17 and TIR18 in Fig. C1. In both cases, the medians of the samples containing clear ice layers are more negative than the medians of the samples not containing clear ice layers, though these medians are relatively close, with a difference of 0.16 ‰ and 0.27 ‰ between the medians, for FK17 and TIR18 respectively. However, FK17 and TIR18 cases are different since the distributions are statistically similar for FK17 $\delta^{18}O$ samples while they are statistically different for TIR18 (Mann-Whitney $U$ test, p < 0.001). This is illustrated by the global shift towards more negative values of the distribution of the samples containing clear ice layers (light burgundy in Fig. C1b). This could indicate crust formation at the surface of colder autumn-winter layers, so care should be taken when interpreting these clear ice layers as "melt layers".

In TIR18 isotopic measurements, 693 samples were identified as containing one or more clear ice layers, for 1700 samples with no identified melt. In FK17, only 260 samples were identified as containing one or more clear ice layers (for 1700 samples with no identified melt). This might be due to different processes taking place at both sites, but less precise logs of the FK17 core while cutting the samples for isotopic measurements cannot be ruled out.

### *Appendix D – Dating procedure*

The following steps have been followed for the dating procedure:

a) *Manual relative dating* supported by the Matchmaker MATLAB® application (Rasmussen et al., 2008), a graphical support to visualize and compare multiple seasonal proxy signals and attribute yearly intervals. To limit the "operator-linked" potential bias, 4 operators individually went through the whole dating process.

As in many other ice core studies, we used the water stable isotopes signal ($\delta^{18}O$) as our primary criterion since it displays a strong seasonal signal. Since the identification of annual layers is sometimes challenging in coastal ice cores, the seasonality of the smoothed ECM and of specific ions (mainly $nssSO_4^{2-}$, $SO_4^{2-}/Na^+$ ratio and MSA) were used as secondary criteria to help deciphering unclear $\delta^{18}O$ patterns. Appendix Fig. D1 illustrates the canvas adopted by all the operators in that respect. In most cases (*case 1* in Fig. D1) the isotopic signal showed successive peaks with a clear interpeak. In those cases, both isotopic peaks were accepted as the indication of the beginning of a new year (1st of January), since maximum temperature, and therefore the related maximum in $\delta^{18}O$ occur in the middle of the austral summer. Mainly two other special cases occurred sporadically: either the rise towards the summer peak presented a shoulder (*special case 2* in Fig. D1), or a sustained plateau was present (*special case 3* in Fig. D1). As shown in the decisional canvas, only individual peaks were accepted when they were also present in the majority of the proxies used as secondary criteria.

b) *Absolute dating tuning* supported by timing of know eruptions and peaks in the $nssSO_4^{2-}$ or ECM records.

Absolute dates of volcanic eruptions were used to validate and fine tune our relative manual dating. Volcanic eruptions are known to temporally increase the $nssSO_4^{2-}$ and ECM signals (typically 1 or 2 years after the eruption). We used peaks higher than $3\sigma$ in the normalized $nssSO_4^{2-}$ and ECM records as potential indicators of volcanic eruptions. We then compared the dates of these specific $>3\sigma$ peaks to known dates of major volcanic eruptions reported within the time window of our cores and we used potential discrepancies to refine our relative dating. The resulting maximum shift was 2-3 years. This is indeed considered as "fine tuning" of an already well-constrained relative manual dating, since, in these coastal areas, peaks in $nssSO_4^{2-}$ and ECM can also result from other non-volcanic processes.

All operators then compared their dating results. The maximum discrepancy between operators was 5 years for the bottom year of the two cores, FK17 and TIR18. Comparison of the individual Matchmaker records were then used by the team to solve the discrepancies and agree on a final manual dating (blue and burgundy lines in Fig. 3).

c) *Automatic dating* using the Straticounter layer counting algorithms (Winstrup et al., 2012), referring to our manual dating to train Straticounter. As stated in Winstrup et al. (2016), the conditions are not optimal in our study cases to run the algorithms, given the high noise level, the interannual variability of the annual layers and the presence of trends in our records. However, this can be partially compensated by using Straticounter with a few tiepoints identified as volcanic eruptions in our manual dating approach. We have chosen 5 tiepoints with
clear $nssSO_4^{2-}$ and/or ECM signature (Pinatubo - 1991, Agung - 1963, Cerro Azul - 1932, Makian - 1861 and Tambora - 1815, Sigl et al., 2013). The resulting automatic dating for our two cores is shown as black lines in Fig. 3.

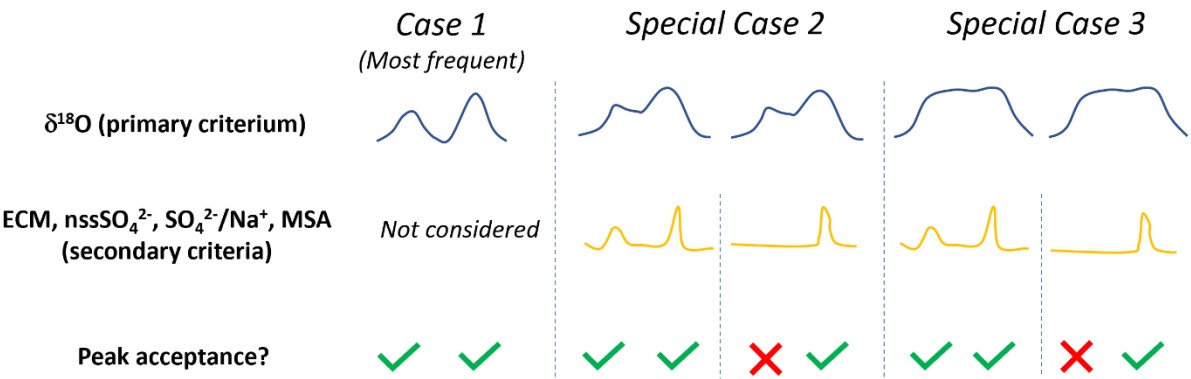

**Figure D1. Canvas adopted by the operators for the relative manual dating process. $\delta^{18}O$ has been chosen as the primary criterion**
**of seasonal cycles. In most cases, $\delta^{18}O$ peaks (taken as January of the year) are separated by clear troughs (case 1). The successive peaks are then considered as successive years. Two main special cases were encountered sporadically, with either a shoulder in the rising or decreasing $\delta^{18}O$ signal (case 2) or a sustained plateau of unusual length (case 3). In those cases, the peaks were accepted when present in the majority of the secondary criterion for seasonal cycles (ECM, $nssSO_4^{2-}$, $SO_4^{2-}/Na^+$, MSA).**

*Appendix E – Conditions of snow accumulation at HIR and LIR*

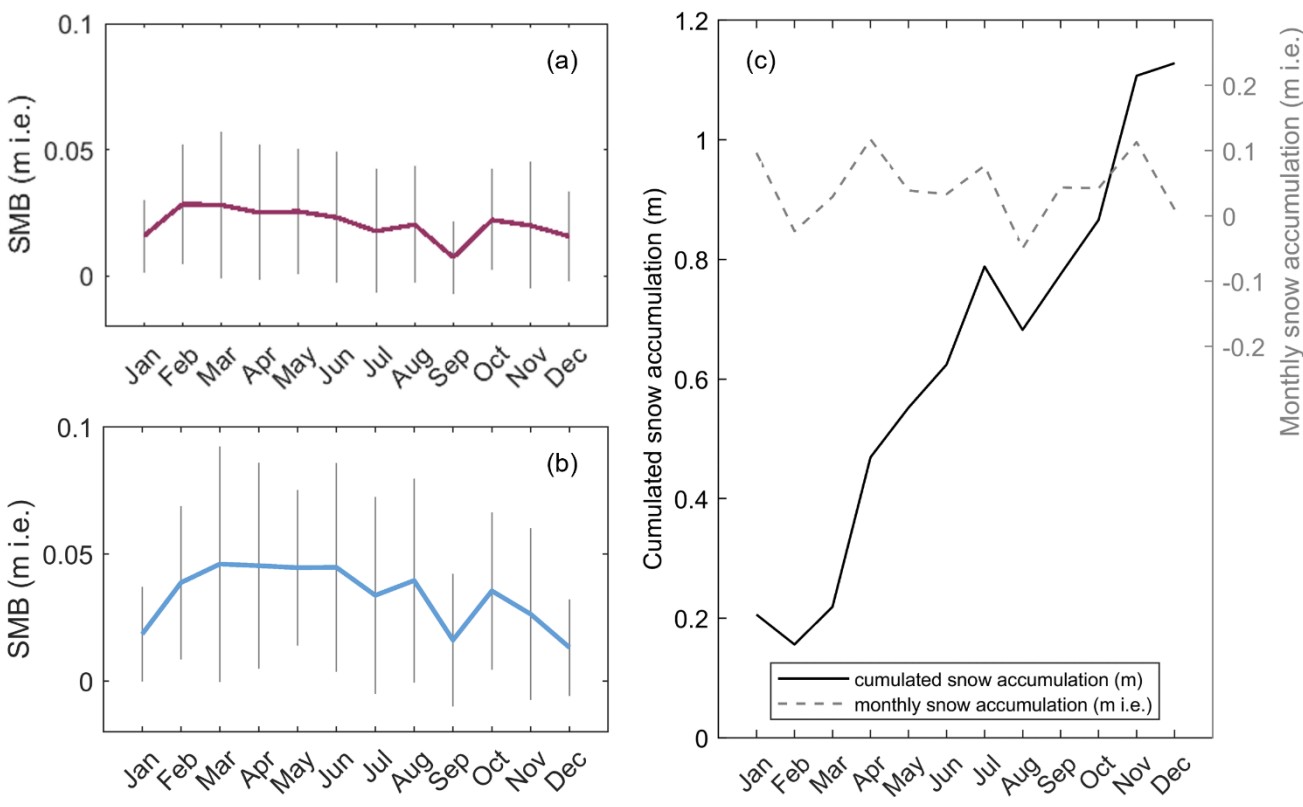

**Figure E1. Conditions of snow accumulation at HIR and LIR. Monthly climatology of RACMO2.3 between 1979 and 2016 at (a) HIR and (b) LIR (based on Lenaerts et al., 2017). The color lines connect the climatology means and the vertical grey bars represent the climatology standard (a measure of the interannual variability). (c) A complete year (2018) of snow accumulation from an AWS located on LIR. The black line corresponds to the monthly cumulated snow height changes and the grey line corresponds to the monthly snow accumulation, calculated from the black line data. The m i.e. scale in (c) is calculated with a reference surface snow density of 430 kg m⁻³.**

*Appendix F – Attribution of volcanic horizons in ECM records*

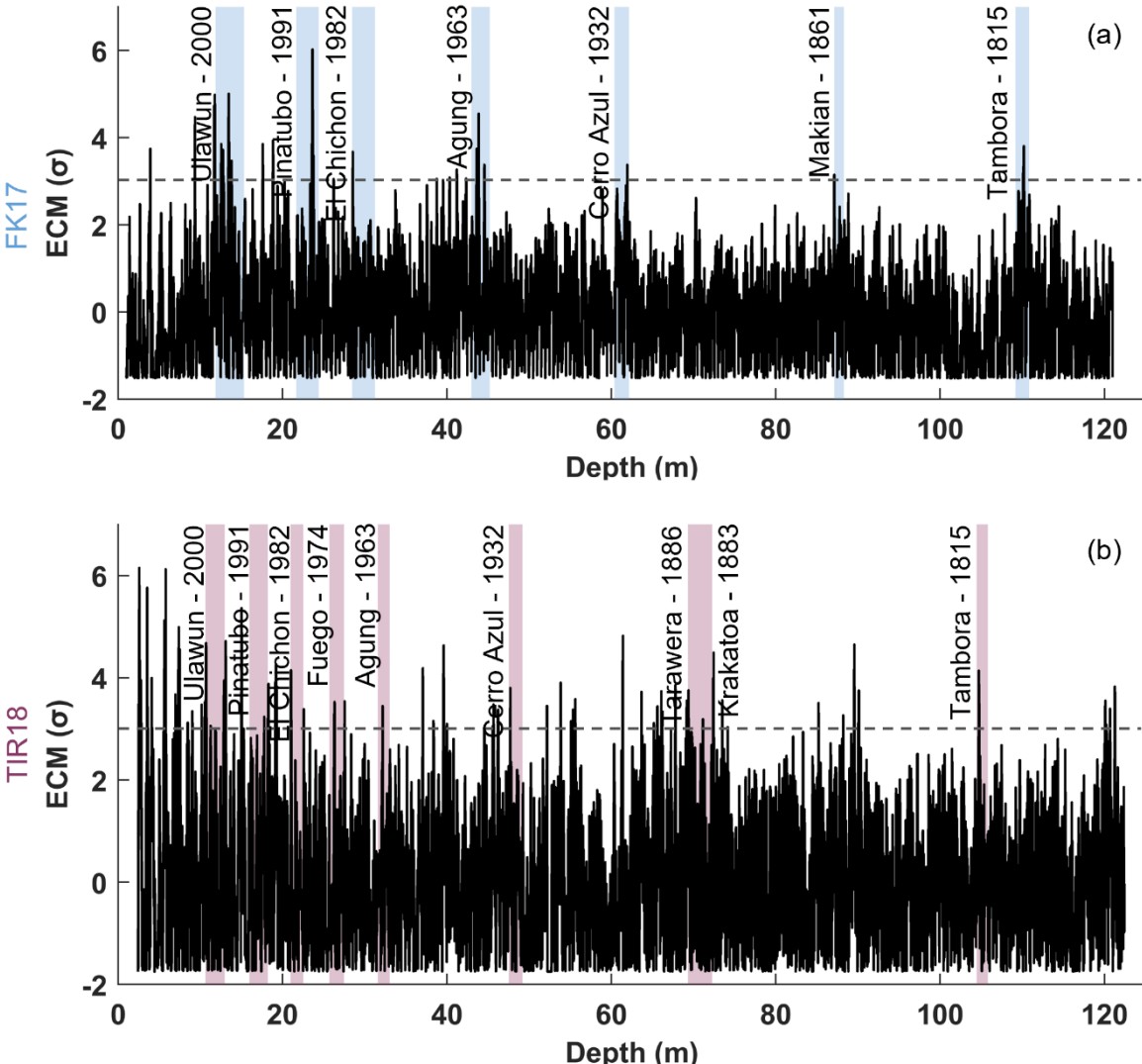

**Figure F1. Normalized ECM records for FK17 (a) and TIR18 (b). The signal (black line) is expressed as a multiple of standard deviation (σ), the 3σ threshold is the dotted horizontal line, and identified volcanic peaks are shown as blue- or burgundy-colored bars, for FK17 and TIR18 respectively. The thicknesses of the color bars are related to the extended period during which the volcanic signal is potentially recorded in the ice core (year of the eruption + 2 years).**

*Appendix G – Comparison with IC12 datasets*

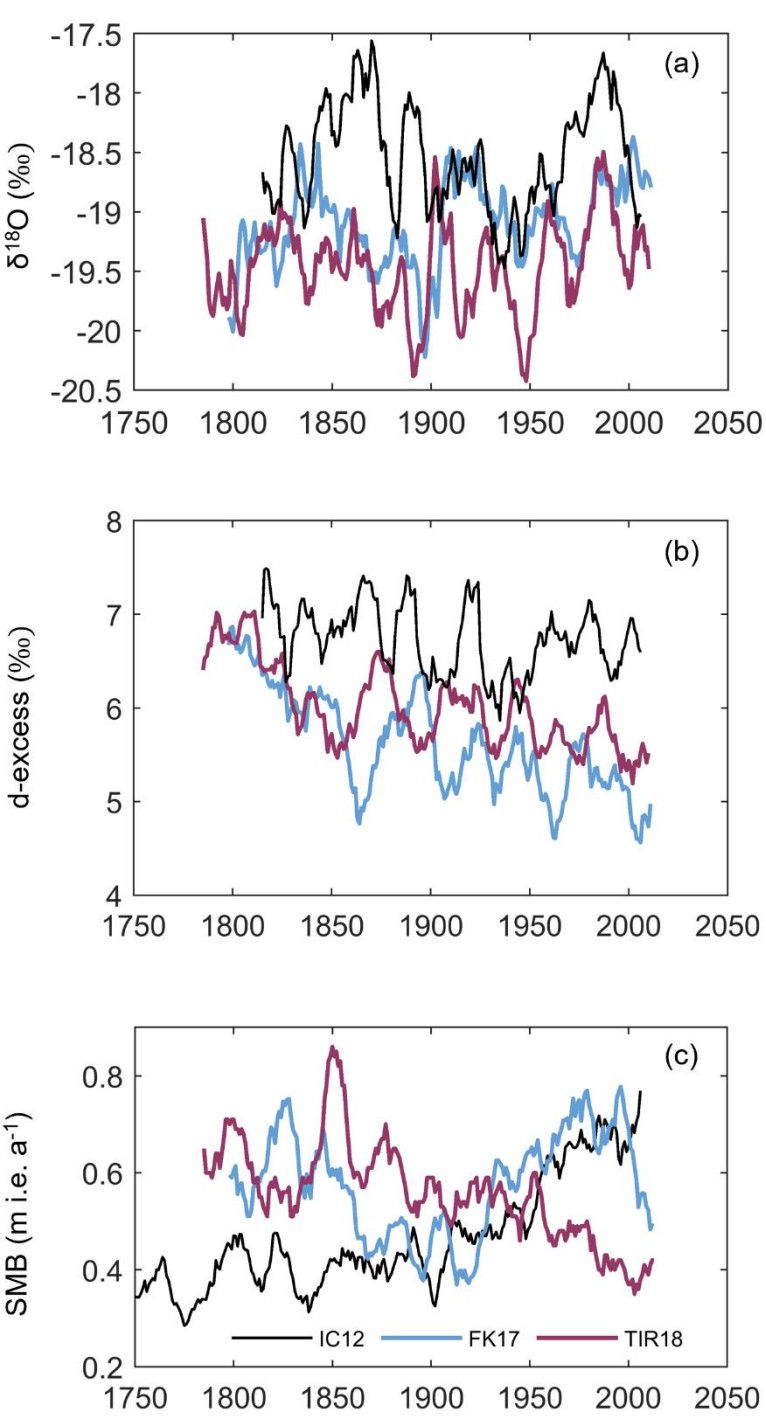

**Figure G1. Comparison with IC12 (Philippe et al., 2016; Philippe and Tison, 2023). Note that these datasets have been smoothed using an 11-yr running mean. IC12 is in black, FK17 in blue and TIR18 in burgundy. (a) Annual mean δ$^{18}$O signal. (b) Annual mean d-excess signal. (c) Surface mass balance corrected for vertical strain rates and expressed in m i.e. a$^{-1}$.**

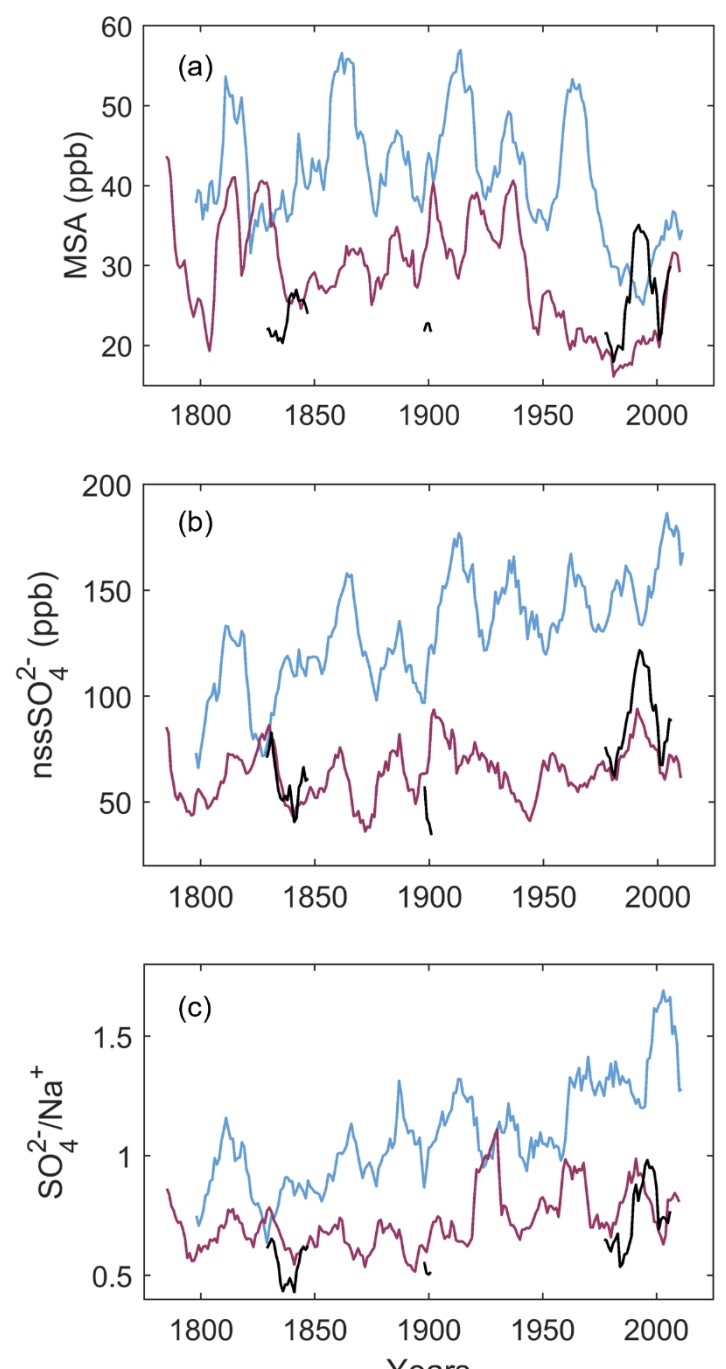

Figure G2. Comparison with IC12 (Philippe et al., 2016; Philippe and Tison, 2023). Note that the IC12 datasets are not complete and have been smoothed using an 11-yr running mean. IC12 is in black, FK17 in blue and TIR18 in burgundy. (a) Annual mean MSA signal. (b) Annual mean nssSO$_4^{2-}$ d-excess signal. (c) Annual mean weight SO$_4^{2-}$/Na$^+$ ratios (sea water: 0.25).

| | $\delta^{18}O$ | | | d-excess | | | MSA | | | nssSO$_4^{2-}$ | | | SO$_4^{2-}$/Na$^+$ | | |
|---|---|---|---|---|---|---|---|---|---|---|---|---|---|---|---|
| | IC12 | FK17 | TIR18 | IC12 | FK17 | TIR18 | IC12 | FK17 | TIR18 | IC12 | FK17 | TIR18 | IC12 | FK17 | TIR18 |
| 1816-1950 | -18.62 | -19.14 | -19.55 | 6.72 | 5.70 | 6.00 | 24.8* | 43.5 | 31.5 | 62.0* | 126.7 | 62.8 | 0.6* | 1.0 | 0.7 |
| 1951-2015 | -18.39 | -18.93 | -19.25 | 6.71 | 5.13 | 5.61 | 25.6* | 36.1 | 22.4 | 87.6* | 149.5 | 68.3 | 0.8* | 1.3 | 0.8 |
| mean | -18.47 | -19.12 | -19.46 | 6.71 | 5.63 | 6.00 | 25.5* | 41.1 | 29.3 | 66.0* | 130.4 | 64.2 | 0.6* | 1.1 | 0.7 |

Table G1. Mean values of $\delta^{18}O$, d-excess, MSA, nssSO$_4^{2-}$ and SO$_4^{2-}$/Na$^+$ of IC12, FK17 and TIR18 for two time periods (1816-1950 and 1951-2015**) and for the entire records. *Note that these IC12 datasets are not continuous. ** The IC12 period is 1951-2011 since the ice core has been drilled in 2012.

| Period (years AD) | SMB (m i.e. a$^{-1}$) | | |
|---|---|---|---|
| | IC12 | FK17 | TIR18 |
| 1816-2011 | 0.52 | 0.58 | 0.55 |
| 1816-1900 | 0.44 | 0.55 | 0.62 |
| 1901-2011 | 0.58 | 0.60 | 0.49 |
| | +32 % | +10 % | -21 % |
| 1816-1961 | 0.47 | 0.54 | 0.59 |
| 1962-2011 | 0.66 | 0.68 | 0.43 |
| | +40 % | +26 % | -26 % |
| 1816-1991 | 0.50 | 0.57 | 0.57 |
| 1992-2011 | 0.68 | 0.66 | 0.40 |
| | +36 % | +16 % | -29 % |

**Table G2. Mean SMB at IC12, FK17 and TIR18 for different time periods (~200, ~100, 50 and 20 years), allowing comparison between the three adjacent ice rises, each ca. 100 km apart. These SMB are corrected for vertical strain rates. The IC12 SMB used is the "oldest estimate" of IC12 as it was the most accurate in Philippe et al., 2016. The % refers, in each case, to the change between the two compared time windows.**

**Author contributions.**

SW, JLT, MI, FF and SEA analyzed the samples. SW, MI and JLT dated the ice cores. SW processed the data derived from the dating. SS and FP collected the radar measurements in the field and processed them. PC provided formation and access to the Picarro instrument. FF provided complementary funding. JLT and FP provided writing support. JLT, MI and FF edited the manuscript. SW wrote the manuscript.

**Competing interests**

The authors declare that they have no conflict of interest.

**Acknowledgments**

The authors wish to thank the International Polar Foundation for logistic support in the field, Dorthe Dahl-Jensen for the formation on the use of the ECM, Mark Curran for the ESTISOL 140 and the two master students for their help during the lab work. We also thank Icefield Instrument Inc. for the design of the ECLIPSE drill and of the ECM and precious help in the field. The authors would also like to thank Nicolas Boucher and Nathalie Focquet, two master's students, for taking part in the analysis and dating of the ice cores, as well as Dr. Brice Van Liefferinge for his critical reading of the manuscript. PC thanks VUB Strategic Research program.

**Financial support**

This work was supported by the Belgian Research Action through Interdisciplinary Networks (BRAIN-be) from the Belgian Science Policy Office (BELSPO) in the framework of the project "East Antarctic surface mass balance in the Anthropocene: observations and multi-scale modelling (Mass2Ant)" (contract no. BR/165/A2/Mass2Ant). The laboratory work also benefited in part from the support of the Fonds Emile Defay (ULB).

**Sarah Wauthy benefited from a Research Fellow grant of the F.R.S.-F.N.R.S.**

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
