# Peer review of "Spatial and temporal variability of environmental proxies from the top 120 m of two ice cores in Dronning Maud Land (East Antarctica)"

_Earth System Science Data, 2023_

## Author Comment (AC1)

**Response to reviewer comments on "**Physico-chemical properties of the top 120 m of two ice cores in Dronning Maud Land (East Antarctica)….**" by Wauthy et al.**

We would like to warmly thank both referees for their very constructive comments which have greatly improved the quality of the manuscript. We provide here below detailed "one to one" response to all their comments and a separate version of the new manuscript with related track changes.

**Referee 1**

Wauthy et al. provided the ice cores data of density, water stable isotopes ($\delta 18O$, $\delta D$ and derived deuterium excess), ions concentrations ($Na^+$, $K^+$, $Mg^+$, $Ca^+$, MSA, $Cl^-$, $SO_4\,2^-$ and $NO_3\,^-$ ), and the continuous electrical conductivity measurement Primary data), as well as the derived data records of age models and surface mass balance from the top 120 m of two ice cores (FK17 and TIR18) drilled on two adjacent ice rises located in coastal Dronning Maud Land, east Antarctica. The authors propose that the datasets presented here offer numerous development possibilities for interpreting the different paleo records and addressing the mechanisms behind the spatial and temporal variability observed at the regional scale in East Antarctica. Since the article is data-based, my comments are restricted to the suitability of data for ESSD and its benefits to the scientific community.

**Major comments**

My primary concern is not providing the complete data for longer climate records of the entire ice core. Here, the authors provide only the ice cores data of the upper 120 m of two adjacent ice rises (Hammarryggen and Lokeryggen) located on the coastal Dronning Maud Land (cDML) along the Princess Ragnhild Coast, East Antarctica. The ice core data profiles provide a time scale to the past 200 to 250 years of ice core data. Several high-resolution ice cores records for similar data profiles have been available from the cDML region for the past 100 to 250 years (e.g. Graf et al., 2002; Naik et al., 2010; Vega et al., 2018; Ejaz et al., 2021; Dey et al., 2022). What is missing from the cDML region is longer climate records from CDML, where westerlies and sea ice extends variabilities were well influenced by the region in the past. The authors claim that the total length of the two ice cores available is 208 m from the Lokeryggen and 268 m from the Hammarryggen ice rises, which maybe provide climate profiles for the past 500 to 700 years data profiles. Hence, I strongly recommend that the authors provide the entire data records for a long time scale, which will be useful to understand past climate variability for a longer time and validate regional/global model results.

Response:

We agree with Referee 1 that it would be ideal to reconstruct past climate profiles on the whole available profiles. However, we can only provide, at this stage, analyses of

the upper 120 m for both ice cores because our laboratory is not equipped with a CFA instrument and the acquisition of the entire datasets (5000 samples multiplied by the number of measured parameters) took more than three years. We believe these results are already worth publishing since they cover the whole Anthropocene transition and document the strong spatial and temporal variability at those spatial and temporal time scales. Aware of the need for longer ice core records, we plan the complete analyses of these ice cores in the coming years.

Regarding the papers mentioned by Referee 1, three of these are already used to support our discussion (i.e. Naik et al., 2010, Vega et al., 2018 and Ejaz et al., 2021). The Graf et al., (2002) paper focuses on the DML plateau and we therefore do not consider it as relevant for our coastal DML focus. As for Dey et al., (2022), we acknowledge the interest of the visual stratigraphy method obtained from line-scan images presented in that paper for the chronology and melt features but this subject is far from our main topic. Note that now we use the results of our detailed "classical" visual inspection of the cores to discuss the potential impact of melt layers in response to one of Referee 2 comments. We would also like to emphasize that most of the mentioned papers either focus on one parameter (Graf et al., 2002 and Naik et al., 2010) or are complementary to another paper (Vega et al., 2018 and Ejaz et al., 2021). Here we present complete datasets (with multiple parameters) for the two ice cores, which have consequences timewise.

As the authors pointed out, accurate dating for shallow cores is tricky. Especially identifying annual layers using water isotopes/major ions was sometimes challenging because of the high noise and background levels due to the coastal location of the ice cores and the post-depositional process. Even the same can be reflected in the $nssSO_4^{2-}$, which identifies volcanic peaks. Hence, I suggest using a proxy that is not influenced by sea supply and does not have many post-depositional effects, such as tritium isotopes (the well-known tritium peak of the largest nuclear bomb testing "Tsar Bomba" in 1961 observed in all compiled tritium records ranges between 1961 and 1962), which were well recorded in the several cDML ice cores. However, the age-depth model and the automatic dating from StratiCounter further improved the uncertainties from the present coastal ice core data.

Response:

Thank you for the suggestion of using tritium isotopes. These are not "standard" measurements and were not used in many other similar studies. They were thus not planned in our analyses schedule. We do not have the possibility to measure these in our laboratory, and the option to do the analyses in another dedicated laboratory would incur considerable delays. However, this will certainly be taken into account when planning for future ice cores projects. As underlined by the referee, the uncertainties are constrained by our age-model and the automatic dating from StratiCounter. Moreover, we have now developed an appendix (Appendix D) to

describe our dating procedure step-by-step and added a full description of the error calculations in section 3.2 as well as error estimates in the figures and tables presenting SMB results.

**Specific comments**

**Section 2.2.2**

Instead of only representing the standard deviation of each ion at line 185, please include a table on precision and accuracy/ any error corrections/SD for all the major ions measurements.

Response:

We replaced the sentence by a table presenting the standard deviation (which measures precision) and added accuracy values (Table 1 in the new version of the manuscript).

**Section 2.4.1**

Since the reconstructed surface mass balance (SMB) from ice cores can be associated with significant uncertainties, especially at the top parts of the ice core, if authors have a shallow radar profile/Radargram from the region, use it for better SMB reconstruction from the ice core and along with other chronological markers/density profile. The same may be included in the text.

Response:

As stated in the paper, Cavitte et al., 2022 showed that our ice cores are representative for a small surface area of the ice rises (~200-500 m radius) but are good representative of the temporal variability. They used shallow radar observations to conclude this. We already mentioned in the original manuscript that: "(SMBs) are systematically 0.08-0.16 m i.e. a$^{-1}$ lower (at the drilling site) than the mean SMB value calculated for the whole ice rise". A potential user of the dataset, interested in estimates of the SMB at the whole ice rise scale can then apply these correction factors to our calculated SMBs. We prefer to keep the values at the core location for comparison to other ice cores. It should also be noted that thse radar SMB estimations rely on dating the ice core.

**Overall comments**

The authors presented new data from two adjacent ice cores from the cDML region, and the data will be helpful for future comparison of paleoclimate data on

regional/global time scales. As mentioned in the above comments, a few additions will improve the data quality/ in the methods and materials sections. Also, additional references/citations to other data sets or missing articles may be incorporated in the comments. The data set is usable in its current format and size, and the article's length is appropriate. All the figures and tables are correct and of high quality. However, the article needs to be improved based on the suggested comments for publication in ESSD.

The abovementioned issues can be addressed with reasonable additions and extra analysis. Hence, I leave it to the Editor to decide whether to recommend modifications and resubmit or leave them all.

**References**

Dey R, Thamban M, Laluraj CM, Mahalinganathan K, Redkar BL, Kumar S, Matsuoka K (2023). Application of visual stratigraphy from line-scan images to constrain chronology and melt features of a firn core from coastal Antarctica. Journal of Glaciology 69(273), 17

Ejaz T, Rahaman W, Laluraj CM, Mahalinganathan K and Thamban M (2021) Sea ice variability and trends in the western Indian Ocean sector of Antarctica during the past two centuries and its response to climatic modes. Journal of Geophysical Research: Atmospheres 126, e2020JD033943. doi: 10.1029/2020JD033943.

Graf, W., Oerter, H., Reinwarth, O., Stichler, W., Wilhelms, F., Miller, H., & Mulvaney, R. (2002). Stable-isotope records from Dronning Maud Land, Antarctica. Annals of Glaciology, 35, 195–201. https://doi.org/10.3189/1727564027818164

Naik SS, Thamban M, Laluraj CM, Redkar BL and Chaturvedi A (2010) A century of climate variability in central Dronning Maud Land, East Antarctica, and its relation to southern annular mode and El Niño[1]Southern oscillation. Journal of Geophysical Research 115(D16), D16102. doi: 10.1029/2009jd013268

Vega, C. P., Isaksson, E., Schlosser, E., Divine, D., Martma, T., Mulvaney, R., Eichler, A., and Schwikowski-Gigar, M.: Variability of sea salts in ice and firn cores from Fimbul Ice Shelf, Dronning Maud Land, Antarctica, The Cryosphere, 12, 1681–1697, https://doi.org/10.5194/tc-12-1681-2018, 2018.

---

## Author Comment (AC2)

**Response to reviewer comments on "**Physico-chemical properties of the top 120 m of two ice cores in Dronning Maud Land (East Antarctica).....**" by Wauthy et al.**

We would like to warmly thank both referees for their very constructive comments which have greatly improved the quality of the manuscript. We provide here below detailed "one to one" response to all their comments and a separate version of the new manuscript with related track changes.

**Referee 2**

In this paper the authors present water isotope data and ion concentrations from the top 120 m of two ice cores drilled at adjacent ice rises in Dronning Maud Land, Antarctica. Both cores were drilled deeper but the whole data sets are not included in this paper. The data from the ice cores presented here cover about 250 years.

The main finding presented in the paper is that the annual records between the two coring sites are quite different despite their geographical closeness. However, on longer time spans the data agree well. These are not new discoveries, but it is nevertheless important to emphasize that short time periods are not necessarily reliable estimates of either SMB or any other climate indicators.

After decades of having the main focus on inland ice cores there has recently been more interest in recovering cores from coastal sites where accumulation is higher, thus with the possibility of obtaining annual records. The coastal ice cores presented here are not the first from this part of Antarctica. But to resolve the complexity of the spatial variability and the impact of various climate induced processes these data sets are important contributions to the understanding of the paleorecords from coastal ice core records from this region.

However, before these datasets can be fully utilized it is necessary with some restructuring and rewriting of the manuscript. Basically, there is some important background information lacking which makes it difficult to evaluate the robustness of the results. Below I have tried to summarize some of my major concerns that I hope that authors will consider for the next version.

**Major comments**

- A major problem with the manuscript is a proper description of the field area and previous work done. As far as I can tell there are two previous papers published including data from these core sites (Kausch et al., 2020 and Cavitte et al., 2022). However, the findings from these studies are not properly integrated in this manuscript. It is not until in "Discussion and perspectives" that this becomes obvious to me. I suggest having a separate "Background" chapter where all this information is included. This should also include information about the meteorology. I find a brief mentioning of an AWS quite far into the manuscript (line 246). Naturally, this information should be supplied at an early stage.
  We developed the section 2.1. to take these suggestions into account:
  - the procedure for the drill site selection is now addressed.

- the use of our datasets to provide preliminary dating for two previous papers (Kausch et al., 2020 and Cavitte et al., 2022) is now mentioned in this 2.1 "Field" section, but we have chosen to keep a more detailed discussion of the findings of these papers in the "Discussion and perspectives" section, in the light of the "Results" section on our own findings.
- a new paragraph now describes briefly the main meteorological information acquired with the AWSs and the temperature and snow accumulation records are shown in a new appendix.

- The dating section needs to be expanded and the error discussed. The authors claim that both cores are annually resolved but no evidence for this is shown. The volcanic chronology is fundamental for the chronology so I would like to see what is described as "the well-defined Tambora marker" (line 318) together with the stable isotope data in these cores to be convinced of the annual resolved dating. As a reader I am left with many questions regarding the chronology. Some examples of my concern are the selection of volcanic eruptions and indications of melt layers.
We agree with the referee that we did not provide enough information on the dating procedure. We expanded the dating section by a complete description of our dating procedure in a new Appendix (Appendix D). This description clarifies our method by explaining it step-by-step and showing the main different cases encountered when dating and how these were dealt with. "The well-defined Tambora marker" (now lines 364-365 in the new version of the manuscript with track changes) is shown in Fig. 2 and Fig. F1. We have considered presenting the raw data used for dating in the form of figures in the Appendices, but we thought it would be inappropriate use of the journal space since the datasets are published online. Finally, Fig. 5 left and central panels show clear seasonality of the species used for dating, which, in our opinion, confirms the annual resolution of the species and thus of the dating.
Melt layers impacts are discussed in our response to another comment here below.

- The SMB is the focus of the paper as I see it and thus there should be more emphasis on the error estimates. All the SMB presented in the tables should also come with the error estimate.
As discussed in the original manuscript, uncertainties on SMBs are important, though tricky to quantify, and not systematically quantified in previous studies. We already gave a theoretical background on the three main types of uncertainties in our section 2.4.1, based on the work of Rupper et al., (2015). We have now built up on it by adding a text on the error estimates in the paragraph presenting the SMB results (section 3.2, lines 340-347 in the new version of the manuscript with track changes), as well as uncertainty ranges in Fig. 4. We also added the errors in the text and tables. For both FK17 and TIR18 records, the average uncertainty for the whole period is ±0.04 m.i.e. for SMB without correction and ±0.05 m.i.e for SMB with vertical strain rate correction.

- In general, the spatial differences between these adjacent ice rises highlight the challenges comparing regional trends using different time periods. I think the manuscript could be stronger emphasizing this. One example: line 65-75 where various records from DML are discussed and it becomes evident that it is

crucial to compare the same time period when discussing data from different sites.

We agree that the same time periods must be compared when discussing the SMB variability, we thus added a sentence (lines 551-553) that emphasize this. The same approach is used when comparing specific time windows in Tables 2, 3, G1 and G2.

line 110. References to the radar measurements are lacking. I assume that the GPR and deep radar measurements were performed prior to the drilling and helped determine the positions of the cores?

Now lines 111-114. The position of the cores had been identified first roughly using REMA and then, more precisely, as the local highest elevation point of the ice rise in the field using GNSS data. This information has been added in the section 2.1.

line 204-205. I wonder what "our previous dating" is referring to? This is one example of where it is not clear if these data have been presented in a previous paper.

Now line 234. The "previous dating" referred to the manual dating (based on relative dating and adjusting with absolute age markers from volcanism) described earlier in the paragraph. We thus replaced the expression by "above-mentioned dating procedure".

line 260-272. As already mentioned, I have issues understanding the process developing a robust the age model. The allocation of peaks as volcanic induced should always be treated carefully both due to spatial coverage and the time lag. Therefore, the double peak of 1809/1815 often just called "Tambora" is an extremely valuable time marker in Antarctica. I would like the authors to add information about this both in the text and in a figure.

As mentioned before, this has been clarified by the extended description of the dating procedure in the Appendix D. The Tambora marker is shown in Fig. 2 and Fig. F1. The "Unknow" volcanic eruption of 1809 was attributed to a peak in each core, but it is smaller than 3σ (2.1σ in FK17 and 2.7σ in TIR18), this is why it is neither showed nor discussed in our paper. Had we chosen a threshold of >2σ as in previous studies, these peaks would have been considered as significant.

line 329. Unclear to me what the expression "globally increasing SMB" refers to here?

Now lines 375-376. It meant that the SMB increased on average on the entire period but is affected by a decrease for the last 20 years. We changed "globally" by "on average" to make it clearer.

line 333. It is curious to see the different seasonality for the MSA peaks in these cores. Then I wonder which seasonality was found in the snow pit data? These data are not part of this paper which is unfortunate.

Now line 380. The shallow cores (no "snow pit data" were performed) were not measured for the major ions (only for water stable isotopes) but it should be underlined that only the first 1 to 3 years were missing from the main cores. The published datasets thus allow to study the seasonality of the first years of the main cores records (see graphs here below): it appears that MSA has clear summer peaks for 6 to 9 years before it starts to show migration towards winter layers. This is expected given the known potential for MSA migration (see lines 531-533 from the manuscript) and the low pace of such a mechanism.

[Figure]

[Figure]

line 410. Could melt layers have contributed to the difference in the stable isotope data?
Now lines 457-458. Melt layers impact on climate proxies have indeed raised an increased interest in the ice core community. We thank the referee for attracting our attention on this. We have used our detailed visual inspection of our cores to track down all potential "melt layers": the melt layers identified in our ice cores were usually very thin (<1 mm) and we do not expect these to have had a significant effect on the measured data having a resolution of 5 cm. However, this information was lacking in our text, we thus added a paragraph describing it (lines 206-214). We also looked for the potential impact of the presence of melt layers on the isotopic frequency distribution. The main findings are now summarized in the text and in an appendix with the figures illustrating the frequency distribution (Appendix C). The median $\delta^{18}O$ differences between samples with and without melt layers (0.16 and 0.27 ‰ for FK17 and TIR18, respectively) are insignificant compared to the seasonal isotopic ranges observed in both cores and also well below the difference observed between the IC12 record and the FK17-TIR18 records that discussed in the text (line 410 - now lines 457-458).

Table 1. The error estimates must be included here. I also wonder why suitable volcanic horizons were not used instead of specific years. That would reduce the error estimates.

Now Table 2. We added the error estimates in the table. We could indeed have chosen volcanic horizons as refence years, but we preferred working on specific time windows (200, 100, 50 and 20 years) to detect potential trends (and acceleration of trends) in different "historic" periods (the end of World War I and industrialisation, the World globalization since the 70's and the last 20 years characterized by important changes worldwide). However, since the datasets are published, it will be possible to consider other time periods (e.g. tied by volcanic eruptions) when comparing our SMB records to other datasets with similar tie points in the future.

Section 4.1. I think that the text about the sources for the ions does not belong here. These are more textbook information.
Section 4.2. We agree with the referee on the "textbook-style" of this paragraph. We however think it is important to briefly remind the sources of the species and their potential as proxies for a wider audience. We have therefore split this, reduced and moved the text in the appropriate paragraphs of the section dedicated to each specific proxy.

**Technical comments**

Title: Should be changed. I think it is too wordy and some of these expression does not quite make sense, i.e. an open window…?
We propose "Spatial and temporal variability of environmental proxies from the top 120 m of two ice cores in Dronning Maud Land (East Antarctica)".

line 100-109. This paragraph belongs in the "Introduction".
Now lines 114-125. Since we developed the section 2.1. to be a more general background section on the previous work done related to field expeditions (drilling site selection, AWS data acquired, resulting publications…), we believe it is coherent to leave the ice rises and ice drilling description there.

line150. Abraham et al. (2013):
Now line 167. This has been modified.

Fig. 1. The choice of colors, font sizes and placement for the names of the ice rises are not well suited. The reference for the Derwael ice rise should not be included on the map.
The font size and placement have been modified to improve the clarity of the figure. The colors have been chosen to be "colorblind-friendly" for the different colorblindness types using the Coblis simulator, with a lighter and a darker color and distinct tones. The reference for the Derwael ice rise has been deleted from the map.

line 96. Philippe et al., (2016)
Still line 96. This has been modified.

line 361. "wealthy datasets" it not a correct English word in this context.
Now line 408. We replaced this term by "information-rich" datasets. We thus changed "the wealth of these datasets" for "the richness of these datasets" (line 90).

Fig. 4. It is difficult to distinguish the lines from each other.
We have divided this figure into 2 parts with the SMB not corrected for vertical strain

rates on the left panels and the SMB corrected for vertical strain rates on the right panels. This allows for better visualization of the data, especially since the SMB uncertainties have been added as colored shadings.

---

## Referee Report (RR1)

This paper presents datasets on accumulation rate, water isotopes and chemistry for the top parts of two ice cores from nearby ice rises. It is a slightly curious paper in that it goes well beyond the material one would expect in a dataset (such as on would find on Pangaea), and yet it doesn't really reach any conclusions. I guess this merely reflects my own uncertainty as to what a data journal accepts – I am happy that the paper allows the authors to describe in detail all the methods by which they obtained their data, including the dating procedure. However the paper appears to promise that papers interpreting the data are still to come – I am a bit mystified what these papers could include other than being repeats of what is here with a little extra speculation. The problem here is that the authors don't really see much they can interpret – the fact that the two cores show different variability and trends means that no large scale conclusions can be drawn, and it is very hard to tell whether recent trends are in any way unusual. Nonetheless, there is a lot of work here, and the data will certainly be useful as food for future compilations of multiple ice cores that may be able to discern underlying trends. I am therefore supportive of the data being published, with the caveat that I don't expect to then be asked to review a later paper that "interprets" the same data with the same methods and the same figures.

The authors have in general answered the main points raised by the initial reviewers. The only substantial issue not addressed is the question of the deeper data that has not yet been obtained: I accept the authors' point that it is already a huge task to produce the data they have. I just ask them, when they do obtain deeper data to ensure that it is easy at that point to find the complete dataset without having to find it twice.

I have a few points where the authors should consider further edits.

Line 33. "The Antarctic ice sheet's future contribution to global sea level rise… is difficult to predict, largely because of the uncertainty and variability of the surface mass balance (SMB)". This is simply not correct – the main reason why the future sea level contribution from Antarctica is hard to predict is well understood to be due to uncertainties in ice dynamics, MISI, MICI, etc as co-author Pattyn has many times written. Please rephrase.

Line 198 and Table 1. I am surprised not to see detection limit as an analytical parameter here. Maybe the concentrations are all well above the DL, in which case just say so to remove doubt.

Fig. 2 – Just a comment that I agree with setting a limit of 3 sigma. Just look how many values are 2 sigma below the mean to see why 2 sigma is not a reliable indicator of a volcanic peak in these noisy coastal cores. Unfortunately this does mean that it's hard to identify clear volcanic peaks – in future work the use of S isotopes to confirm the volcanic nature of some of the peaks used to tie the dating would be worthwhile. (Nothing requested, just a discussion point from me to the authors).

Fig 4 and lines 335-340. The y axis is mislabelled in panels a,c,e: what you are plotting in thise figures is the layer thickness in ice equivalent, NOT the SMB. Only after the correction do you get to SMB. For that reason I actually see no purpose to panel e, nor to giving numbers in the text for "SMB without correction" which is some weird average of a trending layer thickness and is not SMB in any sense. Please re-cast this text and figure y-axes at east.

Fig 5. When you discuss MSA seasonality, we would expect to see it peak in summer near the surface and winter deeper down. You even discuss this in your response to reviewers but I don't think you clarify that here. Wouldn't it be better to show separately the seasonal cycle for the top and then separately for a section deeper down where movement has taken place?

Line 485. "thorough discussion of the processes involved should however be built on fluxes data rather than concentrations". For a site with such high acc rate this is simply not correct. Most chemistry will be wet deposited meaning that the concentration (not the flux) is reflecting the atmospheric concentration. Only for sites with low acc rate, where dry deposition dominates does the flux become important.

Line 500. Regarding the mechanism of MSA movement you may like to quote the excellent paper by Osman et al (Osman, M., Das, S. B., Marchal, O., and Evans, M. J.: Methanesulfonic acid (MSA) migration in polar ice: data synthesis and theory, The Cryosphere, 11, 2439-2462, doi: 10.5194/tc-11-2439-2017, 2017).

---

## Author Response (AR2)

*NOTE to the Editor and Referees*

*Thank you for your inputs in this paper which greatly improved the quality of the final manuscript. Please note that the new modifications are in blue in the "track changes" version. The figures have also been adapted.*

**Referee #2**

**I would like to thank the authors for their careful consideration of reviewer's comments. I appreciated the responses. I think the additional paragraphs and information made it possible for the reader to evaluate the robustness of the results. It seems to me that the authors are fully aware of the weaknesses of their datasets but have tried to make the best of that. This is a valuable data set, nevertheless.**

**At this point I only have a few minor comments. I guess there are more technical issues that I might have missed but in general I think the paper is ready to accept.**

Thank you so much for revisiting the manuscript. It clearly benefited from your first review. Here below our complementary responses to this second round of comments.

**Line 207. How could you see that the potential melt layers less than 1 mm thick were bubble free?**

Line 207. You are correct, this is a misuse of the "bubble free" word, we could clearly see that the largest lenses were bubble free, but this is more difficult to ascertain for the thinnest layers. We therefore changed the words "bubble free layers" for "clear ice layers" throughout the main text (lines 206-212) as well as in the appendix (lines 647-658).

**Line 111. Change to "Pre-selected"**

Still line 111. This has been done.

**Line 480-520. Because nssSo4 and MSA are "closely related" I suggest merging these paragraphs to use the space more efficiently.**

Now lines 480-524. We agree with the referee that these two proxies are closely related. We thus merged the two paragraphs, even though it does not appear in the text with track changes, but it is in the "final" version of the PDF without track changes.

**Line 569. Missing "." after et al. in the two references**

Now line 578. Thank you for seeing this, it has been adjusted.

**Line 712. Change δ18O to correct format**

Now line 726. It has been corrected too, thank you.

**Referee #3**

This paper presents datasets on accumulation rate, water isotopes and chemistry for the top parts of two ice cores from nearby ice rises. It is a slightly curious paper in that it goes well beyond the material one would expect in a dataset (such as on would find on Pangaea), and yet it doesn't really reach any conclusions. I guess this merely reflects my own uncertainty as to what a data journal accepts – I am happy that the paper allows the authors to describe in detail all the methods by which they obtained their data, including the dating procedure. However the paper appears to promise that papers interpreting the data are still to come – I am a bit mystified what these papers could include other than being repeats of what is here with a little extra speculation. The problem here is that the authors don't really see much they can interpret – the fact that the two cores show different variability and trends means that no large scale conclusions can be drawn, and it is very hard to tell whether recent trends are in any way unusual. Nonetheless, there is a lot of work here, and the data will certainly be useful as food for future compilations of multiple ice cores that may be able to discern underlying trends. I am therefore supportive of the data being published, with the caveat that I don't expect to then be asked to review a later paper that "interprets" the same data with the same methods and the same figures.

The authors have in general answered the main points raised by the initial reviewers. The only substantial issue not addressed is the question of the deeper data that has not yet been obtained: I accept the authors' point that it is already a huge task to produce the data they have. I just ask them, when they do obtain deeper data to ensure that it is easy at that point to find the complete dataset without having to find it twice.

Thank you very much for this insight. We will make sure to create a comprehensive, easy-to-find dataset when deeper data will be measured. We agree that we could have left this data paper at the strict stage of the dataset presentation, but we thought it is also part of a dataset paper to show the potential it has for future interpretation. Differently from the reviewer, we believe there are still analyses to be developed on the observed spatial and temporal variability and we are currently working on it (e.g. the role of extreme precipitation events and the use of back-tracking to explain the observed variability). We hope this will not prevent the reviewer from a future revision, if it happens to be so.

I have a few points where the authors should consider further edits.

Line 33. "The Antarctic ice sheet's future contribution to global sea level rise… is difficult to predict, largely because of the uncertainty and variability of the surface mass balance (SMB)". This is simply not correct – the main reason why the future sea level contribution from Antarctica is hard to predict is well understood to be due to uncertainties in ice dynamics, MISI, MICI, etc as co-author Pattyn has many times written. Please rephrase.

Line 35 of the "track change" file. We agree that we have been "over-focusing" on our wn topic. We changed the word "largely" to "partly" to take this remark into account.

Line 198 and Table 1. I am surprised not to see detection limit as an analytical parameter here. Maybe the concentrations are all well above the DL, in which case just say so to remove doubt.

Now line 204. We added a sentence stating that, indeed, the concentrations we measured are well above the detection limit calculated for each ion.

**Fig. 2 – Just a comment that I agree with setting a limit of 3 sigma. Just look how many values are 2 sigma below the mean to see why 2 sigma is not a reliable indicator of a volcanic peak in these noisy coastal cores. Unfortunately this does mean that it's hard to identify clear volcanic peaks – in future work the use of S isotopes to confirm the volcanic nature of some of the peaks used to tie the dating would be worthwhile. (Nothing requested, just a discussion point from me to the authors).**

We agree that 2 sigma is not reliable in these coastal cores. Thank you for the suggestion of using S isotopes, this will certainly be taken into account for future work.

**Fig 4 and lines 335-340. The y axis is mislabelled in panels a,c,e: what you are plotting in thise figures is the layer thickness in ice equivalent, NOT the SMB. Only after the correction do you get to SMB. For that reason I actually see no purpose to panel e, nor to giving numbers in the text for "SMB without correction" which is some weird average of a trending layer thickness and is not SMB in any sense. Please re-cast this text and figure y-axes at east.**

Now lines 349-359. You are right, we should not label the panels (a), (c) and (e) with "SMB", we changed the y-axis for "annual layer thickness" in the figure but also in the text. However, we think it's important to keep the panel (e) since it allows a comparison with panel (f) and clearly shows the impact and necessity of correcting the annual layer thicknesses for vertical strain rates, which is rarely done in the literature presenting longer-term accumulation records from ice cores. We added a sentence to make this clear in our manuscript (lines 353-355 "This comparison highlights the crucial need to correct the annual layer thicknesses with the vertical strain rates when analyzing potential long-term trends in ice core records.").

**Fig 5. When you discuss MSA seasonality, we would expect to see it peak in summer near the surface and winter deeper down. You even discuss this in your response to reviewers but I don't think you clarify that here. Wouldn't it be better to show separately the seasonal cycle for the top and then separately for a section deeper down where movement has taken place?**

To highlight the migration of MSA from peak in summer to peak in winter previously discussed with the reviewers, we have added a few words in our main text (line 536). However, discussing seasonality in detail is not the aim of the dataset paper (see comments above) so we leave it to the reader to have a look at the variation of seasonality based on the available MSA records and age models.

**Line 485. "thorough discussion of the processes involved should however be built on fluxes data rather than concentrations". For a site with such high acc rate this is simply not correct. Most chemistry will be wet deposited meaning that the concentration (not the flux) is reflecting the atmospheric concentration. Only for sites with low acc rate, where dry deposition dominates does the flux become important.**

Now lines 524-526. Thank you for rising this point, we deleted the mentioned section as well as the sentence in the following paragraph.

**Line 500. Regarding the mechanism of MSA movement you may like to quote the excellent paper by Osman et al (Osman, M., Das, S. B., Marchal, O., and Evans, M. J.: Methanesulfonic acid (MSA) migration in polar ice: data synthesis and theory, The Cryosphere, 11, 2439-2462, doi: 10.5194/tc-11-2439-2017, 2017).**

Now lines 535-545. Thank you for suggesting mentioning this very interesting paper and their findings. We added a few sentences summarizing these in our manuscript. The reference has also been added (lines 868-869)

---

## Author Response (AR3)

**Topic editor comments**

We would like to thank the topic editor for his inputs which further improved the manuscript.

**Dear Authors,**

**As previously requested, please use ISO8601 date format or at least unambiguous dates. Specifically, Figure A1b needs a better x-axis, perhaps more similar to most of the other graphics which are not ISO8601 formatted but still easy to read.**

Thank you for noticing the ambiguous dates in Figure A1b and mentioning the helpful ISO8601 standard. Figure A1b has been modified accordingly.

**Additional private note (visible to authors and reviewers only):**

**If you are willing to release your code (analysis, processing, generating figures, etc.), that would be nice. It is usually required. It seems I did not ask for it previously, so it is too late to require it at this point, but it would be nice if you are able to release it (and add a line in the paper)**

We did not use our own codes for the different analyses/processing (except very simple functions for generating figures) but as suggested by the ESSD guidelines, we added a new "Code availability" section after the "Data availability" section where we mention the codes used and where to find them.

**Regards,**

**Ken Mankoff**